# Learning Functional Transduction

**Mathieu Chalvidal**
Capital Fund Management
Paris, France
mathieu.chalvidal@gmail.com

**Thomas Serre**
Carney Institute for Brain Science
Brown University, U.S.
thomas_serre@brown.edu

**Rufin VanRullen**
Centre de Recherche Cerveau & Cognition
CNRS, Universite de Toulouse, France
rufin.vanrullen@cnrs.fr

## Abstract

Research in statistical learning has polarized into two general approaches to perform regression analysis: Transductive methods construct estimates directly based on exemplar data using generic relational principles which might suffer from the curse of dimensionality. Conversely, inductive methods can potentially fit highly complex functions at the cost of compute-intensive solution searches. In this work, we leverage the theory of vector-valued Reproducing Kernel Banach Spaces (RKBS) to propose a hybrid approach: We show that transductive regression systems can be meta-learned with gradient descent to form efficient *in-context* neural approximators of function defined over both finite and infinite-dimensional spaces (operator regression). Once trained, our *Transducer* can almost instantaneously capture new functional relationships and produce original image estimates, given a few pairs of input and output examples. We demonstrate the benefit of our meta-learned transductive approach to model physical systems influenced by varying external factors with little data at a fraction of the usual deep learning training costs for partial differential equations and climate modeling applications.

## 1 Introduction

**Transduction vs. induction** $\diamond$ In statistical learning, transductive inference (Vapnik, 2006) refers to the process of reasoning directly from observed (training) cases to new (testing) cases and contrasts with inductive inference, which amounts to extracting general rules from observed training cases to produce estimates. The former principle powers some of the most successful regression algorithms benefiting from straightforward construction properties, from $k$-Nearest Neighbors (Cover and Hart, 1967) to Support Vector Machines (Boser et al., 1992) or Gaussian Processes (Williams and Rasmussen, 1995). In contrast, deep learning research has mostly endeavored to find inductive solutions, relying on stochastic gradient descent to faithfully encode functional relationships described by large datasets into the weights of a neural network. Although ubiquitous, inductive neural learning with gradient descent is compute-intensive, necessitates large amounts of data, and poorly generalizes outside of the training distribution (Jin et al., 2020) such that a slight modification of the problem might require retraining and cause "catastrophic" forgetting of the previous solution (McCloskey and Cohen, 1989). This may be particularly problematic for real-world applications where data has heterogeneous sources, or only a few examples of the target function are available.

37th Conference on Neural Information Processing Systems (NeurIPS 2023).

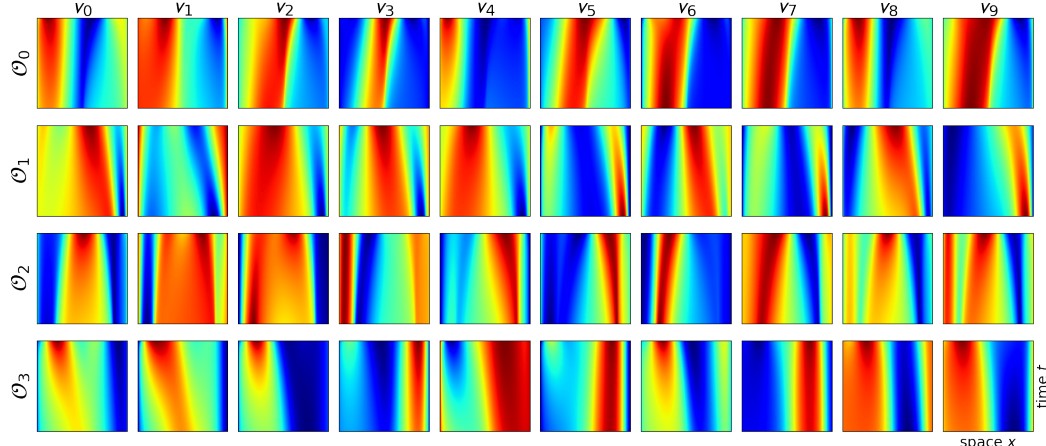

**Figure 1:** Batches of functional images $\mathcal{T}_{\boldsymbol{\theta}}(\mathcal{D}_{\mathcal{O}_i})(\boldsymbol{v}_j) \approx \mathcal{O}_i(\boldsymbol{v}_j) = \boldsymbol{u}_j(x,t) \in C([0,1]^2, \mathbb{R})$ obtained with the same *Transducer* model $\mathcal{T}_{\boldsymbol{\theta}}$ but conditioned, at each row, by a different dataset $(\mathcal{D}_{\mathcal{O}_i})_{i \leqslant 3}$ during feedfoward computation. Each underlying operator $\mathcal{O}_i$ corresponds to a different advection-diffusion-reaction equation (defined in Sec. 5.1) with spatially varying advection, diffusion, and reaction parameters unseen during training, and functions $(\boldsymbol{v}_j)_{j \leqslant 9}$ correspond to initial conditions. **While usual neural regression approaches learn a *single* target function (one row), our model learns to approximate instantaneously an *infinity* of them.**

**Meta-learning functional regression** ⋄ In this work, we meta-learn a regression program in the form of a neural network able to approximate an infinity of functions defined on finite or infinite-dimensional spaces through a transductive formulation of the solution based on the representer theorem. Namely, our model is meta-trained to take as input any dataset $\mathcal{D}_{\mathcal{O}}$ of pairs $(\boldsymbol{v}_i, \mathcal{O}(\boldsymbol{v}_i))_{i \leqslant I}$ of some target function $\mathcal{O}$ together with a query element $\boldsymbol{v}'$ and produces directly an estimate of the image $\mathcal{O}(\boldsymbol{v}')$. After meta-training, our network is able to perform regression of unseen operators $\mathcal{O}'$ from varying dataset sizes in a single feedforward pass, such that our model can be interpreted as performing *in-context functional learning*. In order to build such a model, we leverage the theory of Reproducing Kernel Banach Spaces (RKBS) (Micchelli and Pontil, 2004; Zhang, 2013; Lin et al., 2022) and interpret the Transformer's (Vaswani et al., 2017) attention mechanism as a parametric vector-valued reproducing kernel. While kernel regression might be plagued by the "curse of dimensionality" (Bellman, 1966; Aggarwal et al., 2001), we show that our meta-learning approach can escape this pitfall, allowing, for instance, to perform instantaneous regressions over spaces of operators from a few example points, by building solutions to regression problem instances directly from the general reproducing kernel associated with such spaces.

**Contributions** ⋄ We introduce the *Transducer*, a novel meta-learning approach leveraging reproducing kernel theory and deep learning methods to perform instantaneous regression of an infinity of functions in reproducing kernel spaces.

- Our model learns an implicit regression program able to identify, in a single feedforward pass, elements of specific functional spaces from any corresponding collection of input-output pairs describing the target function. Such ultra-fast regression program, which bypasses the need for gradient-based training, is also general and can be applied to functions either defined on finite dimensional spaces (scalar-valued function spaces) or infinite dimensional spaces (function-valued operator spaces).

- In particular, we demonstrate the flexibility and efficiency of our framework for fitting function-valued operators in two PDEs and one climate modeling problem. We show that our transductive approach allows for better generalization properties of neural operator regression, better precision when relevant data is available, and can be combined with iterative regression schemes that are too expensive for previous inductive approaches, thus holding potential to improve neural operators applicability.

- To the best of our knowledge, our proposal is the first to marry vector-valued RKBS theory with deep meta-learning and might also shed new light on the in-context learning abilities observed in deep attentional architectures.

## 2    Problem formulation

Let $\mathcal{V}$ and $\mathcal{U}$ be two (finite or infinite-dimensional) Banach spaces, respectively referred to as the input and output space, and let $\mathcal{B}$ a Banach space of functions from $\mathcal{V}$ to $\mathcal{U}$. We also note $L(\mathcal{U}, \mathcal{B})$ (resp. $L(\mathcal{U})$) the set of bounded linear operators from $\mathcal{U}$ to $\mathcal{B}$ (resp. to itself). We consider the *meta-learning* problem of creating a function $\mathcal{T}$ able to approximate any functional element $\mathcal{O}$ in the space $\mathcal{B}$ from any finite collection of example pairs $\mathcal{D}_{\mathcal{O}} = \{(\boldsymbol{v}_i, \boldsymbol{u}_i) \mid \boldsymbol{v}_i \in \mathcal{V}, \boldsymbol{u}_i = \mathcal{O}(\boldsymbol{v}_i)\}_{i \leqslant n}$. A prominent approach in statistical learning is *empirical risk minimization* which consists in predefining a class $\tilde{\mathcal{B}} \subset \mathcal{B}$ of computable functions from $\mathcal{V}$ to $\mathcal{U}$ and subsequently selecting a model $\tilde{\mathcal{O}}$ as a minimizer (provided its existence) of a risk function $\mathcal{L} : \mathcal{B} \times \mathcal{D} \mapsto \mathbb{R}$:

$$\mathcal{T}(\mathcal{D}_{\mathcal{O}}) \in \underset{\tilde{\mathcal{O}} \in \tilde{\mathcal{B}}}{\operatorname{argmin}} \mathcal{L}(\tilde{\mathcal{O}}, \mathcal{D}_{\mathcal{O}}) \tag{1}$$

For instance, the procedure consisting in performing gradient-based optimization of objective (1) over a parametric class $\tilde{\mathcal{B}}$ of neural networks defines implicitly such a function $\mathcal{T}$. Fundamentally, this technique works by induction: It captures the statistical regularities of a single map $\mathcal{O}$ into the parameters of the neural network $\tilde{\mathcal{O}}$ such that $\mathcal{D}_{\mathcal{O}}$ is discarded for inference. Recent examples of gradient-based optimization of neural networks for operator regression (i.e when $\mathcal{V}$ and $\mathcal{U}$ are infinite-dimensional) are DeepOnet (Lu et al., 2019) or Fourier Neural Operator (FNO) (Li et al., 2020a). As previously discussed, for every regression problem instance, evaluating $\mathcal{T}$ with these approaches requires a heavy training procedure. Instead, we show in this work that for specific spaces $\mathcal{B}$, we can meta-learn a parametric map $\mathcal{T}_{\boldsymbol{\theta}}$ that transductively approximates (in a certain functional sense) any target function $\mathcal{O} \in \mathcal{B}$ given a corresponding dataset $\mathcal{D}_{\mathcal{O}}$ such that:

$$\forall \boldsymbol{v} \in \mathcal{V}, \ \ \mathcal{T}(\mathcal{D}_{\mathcal{O}})(\boldsymbol{v}) = \mathcal{T}_{\boldsymbol{\theta}}(\boldsymbol{v}_1, \mathcal{O}(\boldsymbol{v}_1), \dots, \boldsymbol{v}_n, \mathcal{O}(\boldsymbol{v}_n), \boldsymbol{v}) \approx \mathcal{O}(\boldsymbol{v}) \tag{2}$$

## 3    Vector-valued Reproducing Kernel Banach Space regression

In order to build $\mathcal{T}_{\boldsymbol{\theta}}$, we leverage the structure of Reproducing Kernel Banach Spaces (RKBS) of functions $\mathcal{B}$ and combine it with the universal approximation abilities of deep networks. As we will see in the experimental section, RKBS are very general spaces occurring in a wide range of machine learning applications. We start by recalling some elements of the theory of vector-valued RKBS developed in Zhang (2013). Namely, we will consider throughout *uniform* Banach spaces $\mathcal{S}$ (such condition guarantees the unicity of a compatible semi-inner product $\langle ., . \rangle_{\mathcal{S}} : \mathcal{S} \times \mathcal{S} \mapsto \mathbb{R}$, i.e. $\forall \boldsymbol{s} \in \mathcal{S}, \langle \boldsymbol{s}, \boldsymbol{s} \rangle_{\mathcal{S}} = \|\boldsymbol{s}\|_{\mathcal{S}}^2$ and allows to build a bijective and isometric dual space $\mathcal{S}^*$).

**Theorem 1** (**Vector-valued RKBS** (Zhang, 2013)). *A $\mathcal{U}$-valued reproducing kernel Banach space $\mathcal{B}$ of functions from $\mathcal{V}$ to $\mathcal{U}$ is a Banach space such that for all $\boldsymbol{v} \in \mathcal{V}$, the point evaluation $\delta_{\boldsymbol{v}} : \mathcal{B} \mapsto \mathcal{U}$ defined as $\delta_{\boldsymbol{v}}(\mathcal{O}) = \mathcal{O}(\boldsymbol{v})$ is continuous. In this case, there exists a unique function $\mathcal{K} : \mathcal{V} \times \mathcal{V} \mapsto L(\mathcal{U})$ such that for all $(\boldsymbol{v}, \boldsymbol{u}) \in \mathcal{V} \times \mathcal{U}$:*

$$\begin{cases} \boldsymbol{v}' \mapsto \mathcal{K}(\boldsymbol{v}, \boldsymbol{v}')(\boldsymbol{u}) \in \mathcal{B} \\ \forall \ \mathcal{O} \in \mathcal{B}, \ \langle \mathcal{O}(\boldsymbol{v}), \boldsymbol{u} \rangle_{\mathcal{U}} = \langle \mathcal{O}, \mathcal{K}(\boldsymbol{v}, .)(\boldsymbol{u}) \rangle_{\mathcal{B}} \\ \forall \ \boldsymbol{v}' \in \mathcal{V}, \ \|\mathcal{K}(\boldsymbol{v}, \boldsymbol{v}')\|_{L(\mathcal{U})} \leqslant \|\delta_{\boldsymbol{v}}\|_{L(\mathcal{B}, \mathcal{U})} \|\delta_{\boldsymbol{v}'}\|_{L(\mathcal{B}, \mathcal{U})} \end{cases} \tag{3}$$

Informally, theorem (1) states that RKBS are spaces sufficiently regular such that the image of *any* element $\mathcal{O}$ at a given point $\boldsymbol{v}$ can be expressed in terms of a unique function $\mathcal{K}$. The latter is hence called the *reproducing kernel* of $\mathcal{B}$ and our goal is to leverage such unicity to build the map $\mathcal{T}_{\boldsymbol{\theta}}$. Let $\mathcal{D}$ be the set of all datasets $\mathcal{D}_{\mathcal{O}}$ previously defined. The following original theorem gives the existence of a solution to our meta-learning problem and relates it to the reproducing kernel.

**Theorem 2** (**RKBS representer map**). *Let $\mathcal{B}$ be a $\mathcal{U}$-valued RKBS from $\mathcal{V}$ to $\mathcal{U}$, if for any dataset $\mathcal{D}_{\mathcal{O}} \in \mathcal{D}$, $\mathcal{L}(., \mathcal{D}_{\mathcal{O}})$ is lower semi-continuous, coercive and bounded below, then there exists a function $\mathcal{T} : \mathcal{D} \mapsto \mathcal{B}$ such that $\mathcal{T}(\mathcal{D}_{\mathcal{O}})$ is a minimizer of equation (1). If $\mathcal{L}$ is of the form $\mathcal{L}(., \mathcal{D}_{\mathcal{O}}) = \tilde{\mathcal{L}} \circ \{\delta_{\boldsymbol{v}_i}\}_{i \leqslant n}$ with $\tilde{\mathcal{L}} : \mathcal{U}^n \mapsto \mathbb{R}$, then the dual $\mathcal{T}(\mathcal{D}_{\mathcal{O}})^*$ is in $\overline{span}\{\mathcal{K}(\boldsymbol{v}_i, .)(\boldsymbol{u})^*, i \leqslant n, \boldsymbol{u} \in \mathcal{U}\}$. Furthermore, if for any $\mathcal{D}_{\mathcal{O}}$, $\mathcal{L}(., \mathcal{D}_{\mathcal{O}})$ is strictly-convex, then $\mathcal{T}$ is unique.*

While theorem (2) provides conditions for the existence of solutions to each regression problem defined by (1), the usual method consisting in solving instance-specific minimization problems

derived from representer theorems characterizations is generally intractable in RKBS for several reasons (non-convexity and infinite-dimensionality of the problem w.r.t to variable $\boldsymbol{u}$, non-additivity of the underlying semi-inner product). Instead, we propose to define image solutions $\mathcal{T}(\mathcal{D}_{\mathcal{O}}) = \sum_{i \leqslant n} \mathcal{K}_{\boldsymbol{\theta}}(\boldsymbol{v}_i, .)(\tilde{\boldsymbol{u}}_i)$ where $\mathcal{K}_{\boldsymbol{\theta}}$ and $(\tilde{\boldsymbol{u}}_i)$ are respectively the learned approximation of the $\mathcal{U}$-valued reproducing kernel $\mathcal{K}$ and a set of functions in $\mathcal{U}$ resulting from a sequence of deep transformations of image examples $(\boldsymbol{u}_i)$ that we define below.

**Transformers attention as a reproducing kernel** $\diamond$ We first need to build $\mathcal{K}$. Several pieces of work have proposed constructions of $\mathcal{K}$ in the context of a non-symmetric and nonpositive semi-definite real-valued kernel (Zhang et al., 2009; Georgiev et al., 2014; Lin et al., 2019; Xu and Ye, 2019). In particular, the exponential key-query function in the popular Transformer model (Vaswani et al., 2017) has been interpreted as a real-valued reproducing kernel $\kappa_{\boldsymbol{\theta}} : \mathcal{V} \times \mathcal{V} \mapsto \mathbb{R}$ in Wright and Gonzalez (2021). We extend below this interpretation to more general vector-valued RKBS:

**Proposition 1** (**Dot-product attention as $\mathcal{U}$-valued reproducing kernel**). *Let $(p_j)_{j \leqslant J}$ a finite sequence of strictly positive integers, let $(A_{\boldsymbol{\theta}}^j)_{j \leqslant J}$ be applications from $\mathcal{V} \times \mathcal{V}$ to $\mathbb{R}$, let $V_{\boldsymbol{\theta}}^j$ be linear applications from $L(\mathcal{U}, \mathbb{R}^{p_j})$ and $W_{\boldsymbol{\theta}}$ a linear application from $L(\prod_{j \leqslant J} \mathbb{R}^{p_j}, \mathcal{U})$, the (multi-head) application $\kappa_{\boldsymbol{\theta}} : \mathcal{V} \times \mathcal{V} \mapsto L(\mathcal{U})$ defined by*

$$\kappa_{\boldsymbol{\theta}}(\boldsymbol{v}, \boldsymbol{v}')(\boldsymbol{u}) \triangleq W_{\boldsymbol{\theta}}\left( \left[..., A_{\boldsymbol{\theta}}^j(\boldsymbol{v}, \boldsymbol{v}') \cdot V_{\boldsymbol{\theta}}^j(\boldsymbol{u}), ...\right]_{j \leqslant J} \right) \tag{4}$$

*is the reproducing kernel of an $\mathcal{U}$-valued RKBS. In particular, if $\mathcal{U} = \mathcal{V} = \mathbb{R}^p$, for $p \in \mathbb{N}^+$ and $A_{\boldsymbol{\theta}}^j = \exp\left(\frac{1}{\tau}(Q_{\boldsymbol{\theta}}^j \boldsymbol{v})^T (K_{\boldsymbol{\theta}}^j \boldsymbol{v}')\right)/\sigma(\boldsymbol{v}, \boldsymbol{v}')$ with $(Q_{\boldsymbol{\theta}}^j, K_{\boldsymbol{\theta}}^j)_{j \leqslant J}$ applications from $L(\mathcal{V}, \mathbb{R}^d)$, $\kappa_{\boldsymbol{\theta}}$ corresponds to the dot-product attention mechanism of Vaswani et al. (2017).*

Note that in (4), the usual softmax normalization of the dot-product attention is included in the linear operations $A_{\boldsymbol{\theta}}^j$ through $\sigma$. We show in the next section how such kernel construction can be leveraged to build the map $\mathcal{T}_{\theta}$ and that several variations of the kernel construction are possible, depending on the target space $\mathcal{B}$ and applications. Contrary to usual kernel methods, our model jointly builds the full reproducing kernel approximation $\mathcal{K}_{\boldsymbol{\theta}}$ and the instance-specific parametrization $(\tilde{\boldsymbol{u}}_i)_{i \leqslant I}$ by integrating the solutions iteratively over several residual kernel transformations. We refer to our system as a *Transducer*, both as a tribute to the Transformer computation mechanism from which it is inspired and by analogy with signal conversion devices.

## 4   The Transducer

**Model definition** $\diamond$ We define $\mathcal{T}_{\boldsymbol{\theta}}$ as the sum of $L$ residual kernel transformations $\{\kappa_{\boldsymbol{\theta}}^\ell\}_{\ell \leqslant L}$ whose expression can be written:

$$\forall \boldsymbol{v} \in \mathcal{V}, \ \ \mathcal{T}_{\boldsymbol{\theta}}(\mathcal{D}_{\mathcal{O}})(\boldsymbol{v}) = \sum_{i \leqslant I} \mathcal{K}_{\boldsymbol{\theta}}(\boldsymbol{v}_i, \boldsymbol{v})(\tilde{\boldsymbol{u}}_i) = \sum_{i \leqslant I} \sum_{\ell \leqslant L} \kappa_{\boldsymbol{\theta}}^\ell(\boldsymbol{v}_i^\ell, \boldsymbol{v}^\ell)(\boldsymbol{u}_i^\ell) \tag{5}$$

where $(\boldsymbol{v}_i^\ell, \boldsymbol{u}_i^\ell)_{i \leqslant n, l \leqslant L}$ and $(\boldsymbol{v}^\ell)_{l \leqslant L}$ refer to sequences of representations starting respectively with $(\boldsymbol{v}_i^1, \boldsymbol{u}_i^1)_{i \leqslant n} = \mathcal{D}_{\mathcal{O}}$, $\boldsymbol{v}^1 = \boldsymbol{v}$ and defined by the following recursive relation:

$$\begin{cases} \boldsymbol{v}_i^{\ell+1} = F_{\boldsymbol{\theta}}^\ell(\boldsymbol{v}_i^\ell), \ \ \boldsymbol{v}^{\ell+1} = F_{\boldsymbol{\theta}}^\ell(\boldsymbol{v}^\ell) \\ \boldsymbol{u}_i^{\ell+1} = \tilde{\boldsymbol{u}}_i^\ell + \sum_j \kappa_{\boldsymbol{\theta}}^\ell(\boldsymbol{v}_j^{\ell+1}, \boldsymbol{v}_i^{\ell+1})(\tilde{\boldsymbol{u}}_j^\ell) \text{ where } \tilde{\boldsymbol{u}}_i^\ell = G_{\boldsymbol{\theta}}^\ell(\boldsymbol{u}_i^\ell) \end{cases} \tag{6}$$

where $(F_{\boldsymbol{\theta}}^\ell, G_{\boldsymbol{\theta}}^\ell)_{\ell \leqslant L}$ correspond to (optional) parametric non-linear residual transformations applied in parallel to representations $(\boldsymbol{v}_i^\ell, \boldsymbol{v}_i^\ell)_{i \leqslant n}$ while $(\kappa_{\boldsymbol{\theta}}^\ell)_{\ell \leqslant L}$ are intermediate kernel transformations of the form $\kappa : \mathcal{V} \times \mathcal{V} \mapsto \mathcal{L}(\mathcal{U})$ such as the one defined in equation (4). Breaking down kernel estimation through this sequential construction allows for iteratively refining the reproducing kernel estimate and approximating on-the-fly the set of solutions $(\tilde{\boldsymbol{u}}_i)_{i \leqslant I}$. We particularly investigate the importance of depth $L$ in the experimental section. Note that equations (5) and (6) allow to handle both varying dataset sizes and efficient parallel inference by building the sequences $(\boldsymbol{v}^\ell)_{\ell \leqslant L}$ with $(\boldsymbol{v}_i^\ell)_{i \leqslant n, \ell \leqslant L}$ in batches and simply masking the unwanted cross-relational features during the kernel operations. All the operations are parallelizable and implemented on GPU-accelerated tensor manipulation libraries such that each regression with $\mathcal{T}_{\boldsymbol{\theta}}$ is orders of magnitude faster than gradient-based regression methods.

**Discretization** ◇ In the case of infinite-dimensional functional input and output spaces $\mathcal{V}$ and $\mathcal{U}$, we can accommodate, for numerical computation purposes, different types of function representations previously proposed for neural operator regression and allowing for evaluation at an arbitrary point of their domain. For instance, output functions $\boldsymbol{u}$ can be defined as a linear combination of learned or hardcoded finite set of functions, as in Lu et al. (2019) and Bhattacharya et al. (2020). We focus instead on a different approach inspired by Fourier Neural Operators (Li et al., 2020a), by applying our model on the $M$ first modes of a fast Fourier transform of functions $(\boldsymbol{v}_i, \boldsymbol{u}_i)_{i \leqslant n}$, and transform back its output, allowing us to work with discrete finite function representations.

**Meta-training** ◇ In order to train $\mathcal{T}_{\boldsymbol{\theta}}$ to approximate a solution for all problems of the form (1), we jointly learn the kernel operations $(\boldsymbol{\kappa}_{\boldsymbol{\theta}}^{\ell})_{\ell \leqslant L}$ as well as transformations $(F_{\boldsymbol{\theta}}^{\ell})_{\ell \leqslant L}$. Let us assume that $\mathcal{L}$ is of the form $\mathcal{L}(\mathcal{O}', \mathcal{D}_{\mathcal{O}}) = \sum_j \tilde{\mathcal{L}}(\mathcal{O}'(\boldsymbol{v}_j), \mathcal{O}(\boldsymbol{v}_j))$, that datasets $\mathcal{D}_{\mathcal{O}}$ are sampled according to a probability distribution $\mathfrak{D}$ over the set of possible example sets with finite cardinality and that a random variable $\mathfrak{T}$ select the indices of each test set $\mathcal{D}_{\mathcal{O}}^{test} = \{(\boldsymbol{v}_i, \boldsymbol{u}_i) \mid (\boldsymbol{v}_j, \boldsymbol{u}_j) \in \mathcal{D}_{\mathcal{O}}, j \in \mathfrak{T}\}$ such that the train set is $\mathcal{D}_{\mathcal{O}}^{train} = \mathcal{D}_{\mathcal{O}} \backslash \mathcal{D}_{\mathcal{O}}^{test}$. Our meta-learning objective is defined as:

$$\mathcal{J}(\boldsymbol{\theta}) = \mathbb{E}_{\mathfrak{D}, \mathfrak{T}} \Big[ \sum_{j \in \mathfrak{T}} \tilde{\mathcal{L}}(\mathcal{T}_{\boldsymbol{\theta}}(\mathcal{D}_{\mathcal{O}}^{train})(\boldsymbol{v}_j), \mathcal{O}(\boldsymbol{v}_j)) \Big] \tag{7}$$

which can be tackled with gradient-based optimization w.r.t parameters $\boldsymbol{\theta}$ provided $\mathcal{L}$ is differentiable (see S.I for details). In order to estimate gradients of (7), we gather a meta-dataset of $M$ operators example sets $(\mathcal{D}_{\mathcal{O}_m})_{m \leqslant M}$ and form, at each training step, a Monte-Carlo estimator over a batch of $k$ datasets from this meta-dataset with random train/test splits $(\mathfrak{T}_k)$. For each dataset in the batch, in order to form outputs $\mathcal{T}_{\boldsymbol{\theta}}(\mathcal{D}_{\mathcal{O}}^{train})(\boldsymbol{v}_j)$ defined by equation (5), we initialize the model sequence in (6) by concatenating $\mathcal{D}_{\mathcal{O}}^{train}$ with $\mathcal{D}_{\mathcal{O}}^{query} = \{(\boldsymbol{v}_i, 0_{\mathcal{U}}) \mid \boldsymbol{v}_i \in \mathcal{D}_{\mathcal{O}}^{test}\}$ and obtain each infered output $\mathcal{T}_{\boldsymbol{\theta}}(\mathcal{D}_{\mathcal{O}}^{train})(\boldsymbol{v}_j)$ as $\sum_{\boldsymbol{v}_i \in \mathcal{D}_{\mathcal{O}}^{train}} \mathcal{K}_{\boldsymbol{\theta}}(\boldsymbol{v}_i, \boldsymbol{v}_j)(\tilde{\boldsymbol{u}}_i)$. Since each regression consists in a single feedforward pass, estimating gradients of the meta-parameters $\boldsymbol{\theta}$ with respect to $\mathcal{L}$ for each batch consists in a single backward pass achieved through automatic differentiation.

# 5 Numerical experiments

In this section, we show empirically that our meta-optimized model is able to approximate any element $\mathcal{O}$ of diverse function spaces $\mathcal{B}$ such as operators defined on scalar and vector-valued function spaces derived from parametric physical systems or regression problems in Euclidean spaces. In all experiments, we use the Adam optimizer (Kingma and Ba, 2014) to train for a fixed number of steps with an initial learning rate gradually halved along training. All the computation is carried on a single Nvidia Titan Xp GPU with 12GB memory. Further details can be found in S.I.

## 5.1 Regression of Advection-Diffusion Reaction PDEs

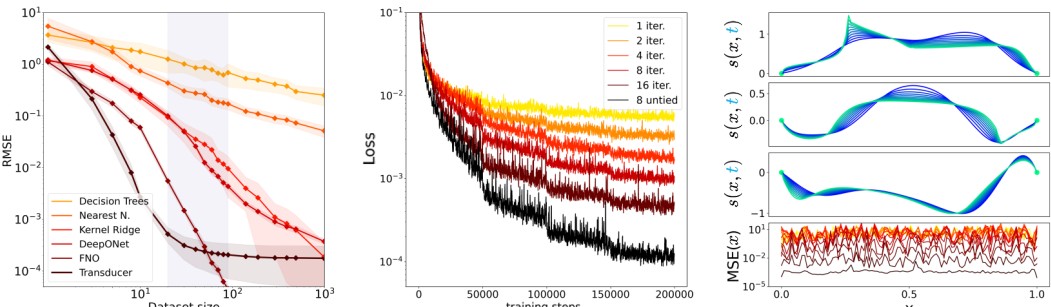

**Figure 2: Left:** RMSEs (and 95% C.I) on unseen operators as a function of the dataset size. The grey area corresponds to dataset cardinalities seen during the *Transducer* meta-training. For comparison, we train baselines from scratch with the corresponding number of examples. **Middle:** Training losses of *Transducers* with different depths. Applying several times the kernel improves performance. Untied weights yield the best performance. **Right:** (*Up*) 3 examples of the evolution of $s(x, t)$ for different ADR equations and (*bottom*) spatial MSEs of intermediate representations $(\boldsymbol{u}^{\ell})$ colored by iteration $\ell$. The decreasing error, consistent with the MSE reduction of deeper models, suggests that network depth allows for progressively refining function estimates.

First, we examine the problem of regressing operators $\mathcal{O}$ associating functions $\boldsymbol{v}$ from $\mathcal{V} \subset C([0,1],\mathbb{R})$ to their solutions $\boldsymbol{u} = \mathcal{O}(\boldsymbol{v}) \subset C([0,1],\mathbb{R})$ with respect to advection-diffusion-reaction equations defined on the domain $\Omega = [0,1] \times [0,t]$ with Dirichlet boundary conditions $\boldsymbol{s}(0,t) = \boldsymbol{s}(1,t) = 0$. We consider the space $\mathcal{B}$ of operators $\mathcal{O}_{(\boldsymbol{\delta},\boldsymbol{\nu},\boldsymbol{k},t)}$ specifically defined by $\boldsymbol{v}(x) = \boldsymbol{s}(x,0)$, $\boldsymbol{u}(x) = \boldsymbol{s}(x,t)$ and $\boldsymbol{s}$ follows an equation depending on unknown random continuous spatially-varying diffusion $\boldsymbol{\delta}(x)$, advection $\boldsymbol{\nu}(x)$, and a scalar reaction term $\boldsymbol{k} \sim \mathcal{U}[0,0.1]$:

$$\partial_t \boldsymbol{s}(x,t) = \underbrace{\nabla \cdot (\boldsymbol{\delta}(x)\nabla_x \boldsymbol{s}(x,t))}_{\text{diffusion}} + \underbrace{\boldsymbol{\nu}(x)\nabla_x \boldsymbol{s}(x,t)}_{\text{advection}} + \underbrace{\boldsymbol{k} \cdot (\boldsymbol{s}(x,t))^2}_{\text{reaction}} \tag{8}$$

Eq. (8) is generic with components arising in many physical systems of interest, leading to various forms of solutions $\boldsymbol{s}(x,t)$. (We show examples for three different operators in figure 2.) Several methods exist for modeling such PDEs, but they require knowledge of the underlying parameters $(\boldsymbol{\delta}, \boldsymbol{\nu}, \boldsymbol{k})$ and often impose constraints on the evaluation point as well as expensive time-marching schemes to recover solutions. Here instead, we assume no *a priori* knowledge of the solution and directly regress each operator $\mathcal{O}$ behavior from the example set $\mathcal{D}_{\mathcal{O}}$.

| Method | RMSE | Time (s) | GFLOPs |
|---|---|---|---|
| FNO | $2.96e^{-4}$ | $1.72e^2$ | $1.68e^2$ |
| DEEPONET | $2.02e^{-2}$ | $7.85e^1$ | $1.54e^2$ |
| FNO-MAML | $1.4e^{-1}$ | $2.10e^0$ | $1.6e^{-1}$ |
| TRANSDUCER | $\mathbf{2.39e^{-4}}$ | $\mathbf{3.10e^{-3}}$ | $\mathbf{1.06e^{-1}}$ |

**Table 1:** RMSE and compute costs of regression over 50 unseen datasets with $n = 50$ examples. Note that DeepONet and FNO are optimized from scratch while the *Transducer* and *FNO-MAML* have been pre-trained. GFLOPs represent the total number of floating point operations for regression.

**Baselines and evaluation** ⋄ We meta-trained our model to regress 500 different operators $\mathcal{O}_{(\boldsymbol{\delta},\boldsymbol{\nu},\boldsymbol{k},1)}$ with $t = 1$ fixed and varying number of examples $n \in [20,100]$ with images evaluated at 100 equally spaced points $(x_k)_{k \in [\![0,100]\!]}$ on the domain $[0,1]$ and meta-tested on a set of 500 operators with new parameters $\boldsymbol{\delta}, \boldsymbol{\nu}, \boldsymbol{k}$ and initial states $\boldsymbol{v}$. Although not directly equivalent to existing approaches, we compared our method with standard regression methods as well as inductive neural operator approximators. We applied standard finite-dimensional regression methods, $K$-Nearest-Neighbors (Fix and Hodges, 1989), Decision Trees (Quinlan, 1986) and Ridge regression with radial basis kernel (Hastie et al., 2009) to each discretized problems $\left(\{\mathcal{O}(\boldsymbol{v}_j)(x_k) = \boldsymbol{u}_j(x_k)\}_{j,k}\right)$ as well as two neural-based operators to each dataset instance: DeepONet (Lu et al., 2021) and FNO

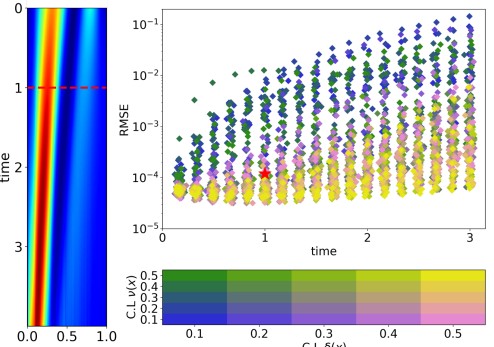

**Figure 3:** Example of *Transducer* regression extrapolation and RMSEs on OOD tasks with $n = 100$ examples. Color code corresponds to different correlation lengths used to generate the random functions $\boldsymbol{\delta}(x)$ and $\boldsymbol{\nu}(x)$. Much of the result remains below 1% error despite never being trained on such operators.

(Li et al., 2020a). Finally, we also tried to meta-learn a parametrization of FNO that could adapt in 100 gradient steps following MAML (Finn et al., 2017) using the same meta-dataset. For all but this approach, an explicit optimization problem is solved before inference in order to fit the target operator. On the other hand, after meta-training of the *Transducer*, which takes only a few minutes to converge, each regression is solved in a single feedforward pass of the network, which is orders of magnitude faster and can be readily applied to new problems (Table 1).

**Results** ⋄ We first verified that our model approximates well unseen operators from the test set (Table 1). We noted that our model learns a non-trivial kernel since the estimation produced with $\ell_2$-Nearest Neighbors remains poor even after $1e^3$ examples. Moreover, since our model can perform inference for varying input dataset sizes, we examined the *Transducer* accuracy when varying the number of examples and found that it learns a converging regression program (Figure 2) which consistently outperforms other instance-specific regression approaches with the exception of FNO when enough data is available ($> 60$). We also found that deeper *Transducer* models with more layers increase kernel approximation accuracy, with untied weights yielding the best performance (figure 2.)

**Extrapolation to OOD tasks** ⋄ We further tested the *Transducer* ability to regress different operators than those seen during meta-training. Specifically, we varied the correlation length (C.L) of the Gaussian processes used to generate functions $\boldsymbol{\delta}(x)$ and $\boldsymbol{\nu}(x)$ and specified a different target time $t' \neq 1$. We showed that the kernel meta-optimized for a solution at $t = 1$ transfers well to these new regression problems and that regression performance degrades gracefully as the target operators behave further away from the training set (figure 3), while inductive solutions do not generalize.

## 5.2 Outliers detection on 2D Burgers' equation

We further show that our regression method can fit operators of vector-valued functions by examining the problem of predicting 2D vector fields defined as a solution of a two-dimensional Burgers' equation with periodic spatial boundary condition on the domain $\Omega = [0,1]^2 \times [0,10]$:

$$\partial_t \boldsymbol{s}(\vec{\boldsymbol{v}},t) = \underbrace{\boldsymbol{\nu}\Delta_{\boldsymbol{v}} \cdot \boldsymbol{s}(\vec{\boldsymbol{v}},t)}_{\text{diffusion}} - \underbrace{\boldsymbol{s}(\vec{\boldsymbol{v}},t)\nabla_{\boldsymbol{x}}\boldsymbol{s}(\vec{\boldsymbol{v}},t)}_{\text{advection}} \tag{9}$$

Here, we condition our model with operators of the form, $\boldsymbol{v}(\vec{\boldsymbol{x}}) = \boldsymbol{s}(\vec{\boldsymbol{x}},t), \boldsymbol{u}(\vec{\boldsymbol{x}}) = \boldsymbol{s}(\vec{\boldsymbol{x}},t')$ such that our model can regress the evolution of the vector field $\vec{\boldsymbol{v}}$ starting at any time, with arbitrary temporal increment $t' - t \leqslant 10$ seconds and varying diffusion coefficient $\boldsymbol{\nu} \in [0.1, 0.5]$. We show in figure (4) and table (2) that our model is able to fit new instances of this problem with unseen parameters $\boldsymbol{\nu}$.

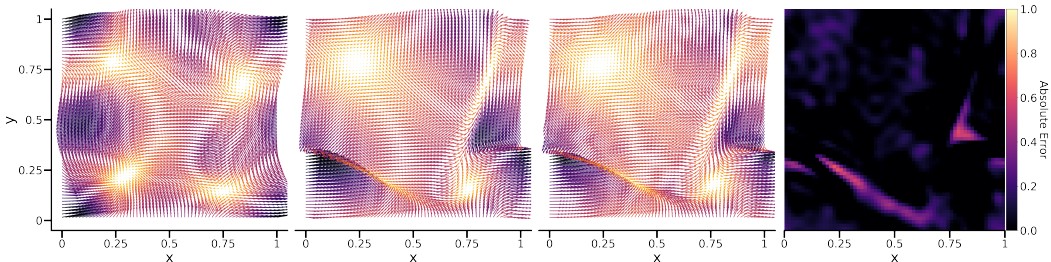

**Figure 4:** Illustrative example of initial ($t = 0$), target ($t = 10$) and *Transducer* estimation of the vector field $s(\vec{\boldsymbol{x}}, t)$ discretized at resolution $64 \times 64$ over the domain $[0,1]^2$ for the Burgers' equation experiment. The last panel represents absolute error to ground truth.

**Fast and differentiable regression** ⋄ Since the fitting operation is orders of magnitude faster than other operator regression approaches as well as fully differentiable, it allows for quickly executing expensive schemes requiring multiple regressions. This can have several applications, from bootstrapping or producing confidence intervals by varying the example set $\mathcal{D}_{\mathcal{O}}^{train}$, or performing inverse problems using Monte-Carlo Markov Chain in the dataset space. We showcase an example of this potential with an outlier detection experiment: We use the *Transducer* to identify outliers of a dataset of Burgers' equation with coefficient $\boldsymbol{\nu}_1$ artificially contaminated with elements from another dataset $\boldsymbol{\nu}_2 > \boldsymbol{\nu}_1$ at 5% level. We identify outliers by estimating RMSEs over 5000 different regressions using random 50 % splits with outliers potentially present in both training and testing sets. This technique takes only a few seconds to estimate while outliers are clearly identified as data points with significantly higher RMSE than the dataset average (figure 5). As a comparison, performing Spectral Clustering (Yu and Shi, 2003) on the FFT of elements ($\boldsymbol{u}_i$) yields very poor precision (table 2)

|  | $t = 5$s | $t = 10$s |
|---|---|---|
| RMSE (test sets) | $2.2e^{-3}$ | $5.9e^{-3}$ |
| Outliers (Pre./Rec.) | 100%/100% | 100%/100% |
| S.C. (Pre./Rec.) | 6%/85% | 7%/85% |

**Table 2 & Figure 5: Left**: Meta-test regression and outlier detection results at two target times. RMSEs on Burgers' equations averaged over 200 different parameter conditions $\boldsymbol{\nu} \in [0.1, 0.5]$ each with 100 train examples. Precision/Recall in outlier detection of the Transducer versus Spectral clustering. **Right**: RMSE distributions of each element in the contaminated dataset over the 5000 regressions. Outliers are clearly identified.

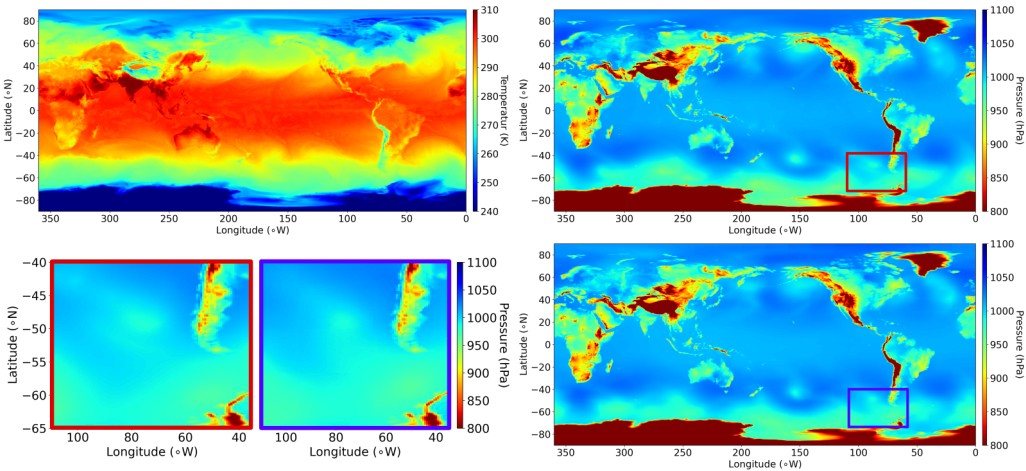

**Figure 6:** *Up* - Illustrative examples of $720 \times 720$ temperature (*left*) and pressure (*right*) fields of the ERA5 dataset. *Bottom* - Estimated pressure field from conditioning the *Transducer* with 15 days data dating 1 week before the target date. Insets show recovered details of the estimation (*blue*) compared with ground truth (*red*).

## 5.3 Climate modeling with seasonal adaptation

One advantage of our approach is the ability to select the data that is most relevant with respect to a certain prediction task and subsequently adapt the model response. For instance, robust and precise prediction of climate variables is difficult because models need to account for seasonal variability and adapt to drifting parameters. Even with a globally large amount of available data, the underlying operator of interest might change over time or be affected by unobserved phenomena. Hence, in order to fully exploit the potential of data-driven methods, being able to capture such variations might greatly help prediction performance on fluctuating and drifting data distributions. In order to illustrate the applicability and scalability of deep transductive learning, we considered the problem of predicting the Earth's surface air pressure solely from the Earth's surface air temperature at a high resolution. Data is taken from the ERA5 reanalysis (Hersbach et al., 2020) publicly made available by the ECMWF, which consists of hourly high-resolution estimates of multiple atmospheric variables from 1979 to the current day. We model pressure estimate on a $720 \times 720$ grid, resulting in a spatial resolution of $0.25° \times 0.5°$, allowing us to capture small features such as local dynamics and geographic relief.

Similar to (Pathak et al., 2022), we modify a ViT backbone to incorporate a kernel transduction layer before every patch attention and compare our model to an unmodified ViT baseline with a matching number of parameters. We additionally compare with a fully transductive Nearest Neighbors approach. In Figure 6 and Table 3, we present results obtained on training a

| METHOD | LWMSE (HPA) | TIME (S) |
|---|---|---|
| NEAREST-NEIGHBORS | 67.326 | 5.915 |
| VIT | 32.826 | **0.053** |
| TRANSDUCER - (P.Y) | 25.293 | 0.192 |
| TRANSDUCER - (P.W) | **22.718** | 0.192 |

**Table 3:** Latitude-weighted mean-square error (in hectopascals) and inference time for the earth surface pressure prediction task.

Transducer with data from 2010 to 2014 and testing it on data from 2016 to 2019. We trained our model by predicting 5 random days sampled from random 20-day windows and present two test configurations: We either condition the *Transducer* with a window centered at the previous year's same date (P.Y) or with a 15 days window lagging by a week (P.W) (see SI for details). Both cases outperform transductive and inductive baselines with fast inference time, confirming that our solution can scale to large problems and be combined with other deep learning modules.

## 5.4 Finite-dimensional case: MNIST-like datasets classification

We finally confirm the generality of our approach in the case of finite-dimensional spaces $\mathcal{U}$ and $\mathcal{V}$ by studying the meta-learning problem presented in Kirsch et al. (2022) which consists in regressing classification functions from the 784-dimensional space of MNIST-like images (LeCun

| METHOD | MNIST | FASHIONMNIST | KMNIST |
|---|---|---|---|
| MAML (FINN ET AL., 2017) | 53.71% | 48.44% | 36.33% |
| VSML (KIRSCH ET AL., 2021) | 79.04% | 68.49% | 54.69% |
| GPICL (KIRSCH ET AL., 2022) | 73.70 % | 62.24% | 53.39% |
| TRANSDUCER | **81.83**% | **69.85**% | **60.64**% |

**Figure 7:** Comparison of meta-test accuracies of MNIST-like datasets classification tasks. Meta-learning models are trained on transformations of MNIST and are meta-tested on original MNIST, FashionMNIST and KMNIST .

and Cortes, 2010) to a 10-dimensional space of one-hot class encoding (i.e functions considered are $\mathcal{O} : [0,1]^{784} \mapsto [0,1]^{10}$). We meta-train a 2-layer Transducer to classify consistently pixel-permuted and class-permuted versions of MNIST. We then meta-test the Transducer to classify the unpermuted MNIST dataset and how the regression map transfer to Fashion-MNIST (Xiao et al., 2017) and KMNIST (Clanuwat et al., 2018). We show in Table 7 that the Transducer outperforms previous meta-learning approaches on both the original MNIST classification task as well as transfer better on Fashion MNIST and K-MNIST classfication tasks.

# 6 Related work

**Transductive Machine learning** ⋄ Principles of transductive statistical estimation have been formally described in Gammerman et al. (1998); Vapnik (1999). Algorithms relying on relational structures between data points such as $K$-nearest neighbors (Cover and Hart, 1967) and kernel methods (Nadaraya, 1964; Watson, 1964) build estimates by weighing examples with respect to a certain metric space. Further, the "kernel trick" allows to embed possibly infinite-dimensional features (Ferraty and Vieu, 2006) into finite Gram matrix representations that are also well-suited for multi-task regression (Evgeniou et al., 2005; Caponnetto et al., 2008). Distinctively, Gaussian processes regression (Williams and Rasmussen, 1995) combines transduction with Bayesian modeling to estimate a posterior distribution over possible functions. These techniques might suffer from the so-called "curse of dimensionality": with growing dimensionality, the density of exemplar point diminishes, which increases estimators' variance. More recent work combining deep learning with transductive inference has shown promising results even in high-dimensional spaces for few-shot learning (Snell et al., 2017; Sung et al., 2018) or sequence modeling (Jaitly et al., 2015), but the vast majority of neural networks still remain purely inductive.

**Neural operator learning** ⋄ The universal approximation abilities of neural networks have been generalized to infinite-dimensional function spaces: Chen and Chen (1995) showed that finite neural parametrization can approximate well infinite-dimensional operators. More recent work using neural networks to perform operator regression has shown strong results (Lu et al., 2019), especially when mixed with tools from functional analysis and physics (Raissi et al., 2017; Li et al., 2020a; Gupta et al., 2021; Li et al., 2020b; Nelsen and Stuart, 2021; Wang et al., 2021; Roberts et al., 2021) and constitutes a booming research direction in particular for physical applications (Goswami et al., 2022; Pathak et al., 2022; Vinuesa and Brunton, 2022; Wen et al., 2022; Pickering et al., 2022). Recently, the Transformer's attentional computation has been interpreted as a Petrov-Galerkin projection (Cao, 2021) or through Reproducing Kernel Hilbert Space theory (Kissas et al., 2022) for building such neural operators, but these perspectives apply attention to fit a single target operator.

**Meta-learning and in-context learning** ⋄ Promising work towards more general and adaptable machines has consisted in automatically "learning to learn" or meta-learning programs (Schmidhuber et al., 1997; Vilalta and Drissi, 2002), by either explicitly treating gradient descent as an optimizable object (Finn et al., 2017), modeling an optimizer as a black-box autoregressive model (Ravi and Larochelle, 2017) or informing sequential strategies via memorization (Santoro et al., 2016; Ortega et al., 2019) More recently, converging findings in various domains from reinforcement learning (Mishra et al., 2018; Laskin et al., 2022), natural language processing (Brown et al., 2020; Xie et al., 2021; Olsson et al., 2022) and functional regression (Garg et al., 2022) have established the ability of set-based attentional computation in the Transformer (Vaswani et al., 2017) for *in-context* learning by flexibly extracting functional relationships and performing dynamic association such as linguistic analogy or few-shot behavioral imitation. We show that the theory of RKBS can help interpret such property and extends it to function-valued operators regression.

# 7 Discussion

We proposed a novel transductive model combining kernel methods and neural networks that is capable of performing regression over entire function spaces. We based our model on the theory of vector-valued Reproducing Kernel Banach Spaces and showcased several instances where it learns a regression program able, in a single feedforward pass, to reach performance levels that match or outperform previous instance-specific neural operators or meta-learning systems. Our approach holds potential to create programs flexibly specified by data and able to model entire families of complex physical systems, with particular applications in functional hypothesis testing, dataset curation or fast ensemble learning. However, one limitation is that our model relies on meta-training, which requires collecting a sufficiently diverse meta-dataset to explore the kernel space. In future work, we plan to investigate methods such as synthetic augmentation to reduce the costs of meta-training.

# 8 Acknowledgements

This work was funded by ANR-3IA Artificial and Natural Intelligence Toulouse Institute (ANR-19-PI3A0004) ONR (N00014-19- 1-2029), NSF (IIS-1912280 and EAR-1925481), DARPA (D19AC00015) and NIH/NINDS (R21 NS 112743). Additional support provided by the Carney Institute for Brain Science and the Center for Computation and Visualization (CCV) and the NIH Office of the Director grant S10OD025181.

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
