# Learning Functional Transduction: S.I.

## Contents

## S.1   Theoretical analysis

We propose below the proofs of the results presented in the main text. Most of the arguments are adapted from the development proposed in (Zhang, 2013) which goes beyond real or complex-valued RKBS developed in (Zhang et al., 2009; Song et al., 2013) to develop the notion of *vector-valued* RKBS. In addition, we note that assumptions regarding the properties of the RKBS of interests such as uniform Fréchet differentiability and uniform convexity have been further relaxed in other works (Xu and Ye, 2019; Lin et al., 2022) but are here sufficient for our discussion since they guarantee the unicity of a semi-inner product $\langle .,.\rangle_{\mathcal{B}}$ compatible with the norm $||.||_{\mathcal{B}}$ (Giles, 1967).

### S.1.1   Theoretical results

**Theorem 1** $\diamond$ Theorem 1 gathers for the sake of compactness the definition of a vector-valued reproducing kernel Banach space with the properties of existence and unicity of the kernel $\mathcal{K}$.

*Proof.* For any $\boldsymbol{v} \in \mathcal{V}$ and $\boldsymbol{u} \in \mathcal{U}$, the mapping $\mathcal{O} \mapsto \langle \mathcal{O}(\boldsymbol{v}), \boldsymbol{u}\rangle_{\mathcal{U}}$ is a bounded linear form in $\mathcal{L}(\mathcal{B})$. By Theorem 7 of Giles (1967), we have the bijectivity of the duality mapping in $\mathcal{U}$, hence there exists a unique element $\mathcal{K}_{\boldsymbol{v},\boldsymbol{u}} \in \mathcal{B}$ such that:

$$\langle \mathcal{O}(\boldsymbol{v}), \boldsymbol{u}\rangle_{\mathcal{U}} = \langle \mathcal{O}, \mathcal{K}_{\boldsymbol{v},\boldsymbol{u}}\rangle_{\mathcal{B}} \tag{1}$$

Hence, this defines a unique function $\mathcal{K} : \mathcal{V} \times \mathcal{V} \mapsto \mathcal{L}(\mathcal{U})$ such that:

$$\forall (\boldsymbol{v}, \boldsymbol{v}') \in \mathcal{V}^2, \ \forall \boldsymbol{u} \in \mathcal{U}, \quad \mathcal{K}(\boldsymbol{v}, \boldsymbol{v}')(\boldsymbol{u}) = \mathcal{K}_{\boldsymbol{v},\boldsymbol{u}}(\boldsymbol{v}') \tag{2}$$

By construction $\mathcal{K}$ is unique, furthermore we have that (i) the functional $\boldsymbol{v}' \mapsto \mathcal{K}(\boldsymbol{v}, \boldsymbol{v}')$ is an element of $\mathcal{B}$ (ii) it verifies the reproducing relation $\forall (\boldsymbol{v}, \boldsymbol{u}), \langle \mathcal{O}(\boldsymbol{v}), \boldsymbol{u}\rangle_{\mathcal{U}} = \langle \mathcal{O}, \mathcal{K}(\boldsymbol{v}, .)(\boldsymbol{u})\rangle_{\mathcal{B}}$. Finally, property (iii) follows from the following bound on the norm of $\boldsymbol{v}' \mapsto \mathcal{K}(\boldsymbol{v}, .)(\boldsymbol{u})$:

$$||\mathcal{K}(\boldsymbol{v}, .)(\boldsymbol{u})||_{\mathcal{B}} \leqslant \sup_{\mathcal{O} \in \mathcal{B}, ||\mathcal{O}||_{\mathcal{B}} \leqslant 1} |\langle \mathcal{O}, \mathcal{K}(\boldsymbol{v}, .)(\boldsymbol{u})\rangle_{\mathcal{B}}| = \sup_{\mathcal{O} \in \mathcal{B}, ||\mathcal{O}||_{\mathcal{B}} \leqslant 1} |\langle \mathcal{O}(\boldsymbol{v}), \boldsymbol{u}\rangle_{\mathcal{U}}| \tag{3}$$

Further, we have by continuity of the point evaluation $\delta_{\boldsymbol{v}} : \mathcal{O} \mapsto \mathcal{O}(\boldsymbol{v})$ that:

$$|\langle \mathcal{O}(\boldsymbol{v}), \boldsymbol{u} \rangle_{\mathcal{U}}| \leqslant ||\delta_{\boldsymbol{v}}||_{\mathcal{L}(\mathcal{B}, \mathcal{U})} \cdot ||\boldsymbol{v}||_{\mathcal{U}} \tag{4}$$

Combining (3) and (4) allows to write:

$$||\mathcal{K}(\boldsymbol{v}, \boldsymbol{v}')(\boldsymbol{u})||_{\mathcal{B}} \leqslant ||\delta_{\boldsymbol{v}'}||_{\mathcal{L}(\mathcal{B}, \mathcal{U})} \cdot ||\mathcal{K}(\boldsymbol{v}, .)(\boldsymbol{u})||_{\mathcal{B}} \leqslant ||\delta_{\boldsymbol{v}'}||_{\mathcal{L}(\mathcal{B}, \mathcal{U})} \cdot ||\delta_{\boldsymbol{v}}||_{\mathcal{L}(\mathcal{B}, \mathcal{U})} \cdot ||\boldsymbol{u}||_{\mathcal{U}} \tag{5}$$

Observing that in particular for all $\boldsymbol{u} \in \mathcal{U} \backslash 0_{\mathcal{U}}$:

$$\frac{||\mathcal{K}(\boldsymbol{v}, \boldsymbol{v}')(\boldsymbol{u})||_{\mathcal{B}}}{||\boldsymbol{u}||_{\mathcal{U}}} \leqslant ||\delta_{\boldsymbol{v}'}||_{\mathcal{L}(\mathcal{B}, \mathcal{U})} \cdot ||\delta_{\boldsymbol{v}}||_{\mathcal{L}(\mathcal{B}, \mathcal{U})} \tag{6}$$

concludes the proof. $\qquad \square$

**Theorem 2** $\diamond$ We first show the existence of a solution for any problem of the form (**??**) and then characterize the solution in terms of the data points.

*Proof.* We first show the existence of the map $\mathcal{T} : \mathcal{D} \mapsto \mathcal{B}$. Let us take $\mathcal{D} \in \boldsymbol{\mathcal{D}}$, by assumption the function $\mathcal{L}_{\mathcal{D}} : \tilde{\mathcal{O}} \mapsto \mathcal{L}(\mathcal{O}, \mathcal{D})$ is weakly-lower semi-continuous, coercive and bounded below. Let us take a sequence $(\mathcal{O}_k)_{k \in \mathbb{N}}$ of elements in $\mathcal{B}$ such that $\mathcal{L}_{\mathcal{D}}(\mathcal{O}_k) \to \mathfrak{L} = \inf_{\mathcal{O} \in \mathcal{B}} \mathcal{L}_{\mathcal{D}}(\mathcal{O})$. Since $\mathcal{L}_{\mathcal{D}}$ is coercive, the sequence is bounded in $\mathcal{B}$, so there is a weakly-convergent subsequence $(\mathcal{O}_{k_i})$ such that $(\mathcal{O}_{k_i}) \to \mathcal{O}_0$. Finally, by property of weakly-lower semi-continuity, we have that $\mathfrak{L} \leqslant \mathcal{L}_{\mathcal{D}}(\mathcal{O}_0) \leqslant \lim \inf \mathcal{L}_{\mathcal{D}}(\mathcal{O}_k)$ which shows that for any $\mathcal{D}$, there exists a minimizer of $\mathcal{L}_{\mathcal{D}}$.

We now turn to the characterization of the solution $\mathcal{O}_0$ when we have that $\mathcal{L}_{\mathcal{D}} = \tilde{\mathcal{L}} \circ \{\delta_{\boldsymbol{v}_i}\}_{i \leqslant n}$ with $\tilde{\mathcal{L}} : \mathcal{U}^n \mapsto \mathbb{R}$. This assumption allows to exhibit a characterization of the solution in terms of annihilator and pre-annihilitors in $\mathcal{B}$ as in previous work (Zhang, 2013; Xu and Ye, 2019). Let us consider the set $S = \{\mathcal{O} \in \mathcal{B}, \mathcal{O}(\boldsymbol{v}_i) = \boldsymbol{u}_i, i \leqslant I\}$. It is clearly a closed convex subset of $\mathcal{B}$. Since $\mathcal{B}$ is uniformly convex, the problem

$$\inf\{||\mathcal{O}||_{\mathcal{B}}, \mathcal{O} \in S\} \tag{7}$$

admits a best approximation in $S$ (**?**). Furthermore, $\mathcal{O}_0$ is the minimizer of (7) if and only if for all $\mathcal{O} \in S_0 = \{\mathcal{O} \in \mathcal{B}, \mathcal{O}(\boldsymbol{v}_i) = 0_{\mathcal{U}}, i \leqslant I\}$, we have:

$$||\mathcal{O} + \mathcal{O}_0||_{\mathcal{B}} \geqslant ||\mathcal{O}_0||_{\mathcal{B}} \tag{8}$$

which by Giles (1967) is equivalent to $\mathcal{O}_0 \in (S_0)^{\perp}$. Finally, we note that $\mathcal{O} \in S_0$ if and only if

$$\langle \mathcal{O}, \mathcal{K}(\boldsymbol{v}_j, .)(\boldsymbol{u}) \rangle_{\mathcal{B}} = \langle \mathcal{O}(\boldsymbol{v}_j), \boldsymbol{u} \rangle_{\mathcal{U}} = 0, \quad \forall j \leqslant n, \ \forall \boldsymbol{u} \in \mathcal{U} \tag{9}$$

which allows us to say that

$$\mathcal{O} \in {}^{\perp}\big\{(\mathcal{K}(\boldsymbol{v}_j, .)(\boldsymbol{u}))^*, j \leqslant n, \boldsymbol{u} \in \mathcal{U}\big\} \tag{10}$$

Finally, we obtain the following characterization: $\mathcal{O} \in \big({}^{\perp}\{(\mathcal{K}(\boldsymbol{v}_j, .)(\boldsymbol{u}))^*, j \leqslant n, \boldsymbol{u} \in \mathcal{U}\}\big)^{\perp}$. Since $\mathcal{B}$ is reflexive, we have further that $\forall S \subset \mathcal{B}, ({}^{\perp}S)^{\perp} = \overline{span}S$, which concludes the proof for the characterization of $\mathcal{T}(\mathcal{D})$.

Finally, if for all $\mathcal{D}$, the function $\mathcal{L}_{\mathcal{D}}$ is strictly-convex, then it guarantees the unicity of a minimizer over $\mathcal{B}$ for every problem, which in turn defines an unique map $\mathcal{T}$. $\qquad \square$

**Proposition 1** $\diamond$ The result is direct by considering the feature map characterization of vector-valued RKBS (Corollary 3.2 of Zhang (2013)) that we recall hereafter: We first define for any linear operator $T \in \mathcal{L}(\mathcal{S}_1, \mathcal{S}_2)$ between two Banach spaces $\mathcal{S}_1, \mathcal{S}_2$, the generalized adjoint $T^{\dagger} \in \mathcal{L}(\mathcal{S}_2, \mathcal{S}_1)$ as the application verifying $\langle Ts, s' \rangle_{\mathcal{S}_1} = \langle s, T^{\dagger}s' \rangle_{\mathcal{S}_2}$ for all $(s, s') \in \mathcal{S}_1 \times \mathcal{S}_2$.

Let $\mathcal{F}$ be a uniform Banach space and $\Phi : \mathcal{V} \mapsto \mathcal{L}(\mathcal{F}, \mathcal{U})$ a feature map such that:

$$\forall (\boldsymbol{v}, \boldsymbol{v}') \in \mathcal{V}^2, \quad \Phi(\boldsymbol{v}')(\Phi^{\dagger}(\boldsymbol{v})) = \mathcal{K}(\boldsymbol{v}, \boldsymbol{v}') \tag{11}$$

$$\overline{span}\{(\Phi^{\dagger}(\boldsymbol{v})(\boldsymbol{u}))^*, \boldsymbol{v} \in \mathcal{V}, \boldsymbol{u} \in \mathcal{U}\} = \mathcal{F}^* \tag{12}$$

with $\Phi^{\dagger} : \mathcal{V} \mapsto \mathcal{L}(\mathcal{U}, \mathcal{F})$ is defined by: $\forall \boldsymbol{v}, \ \Phi^{\dagger}(\boldsymbol{v}) = (\Phi(\boldsymbol{v}))^{\dagger}$. then the vector space $\tilde{\mathcal{B}} = \{\Phi(.)(\boldsymbol{w}) | \boldsymbol{w} \in \mathcal{F}\}$ endowed with the norm $||\Phi(.)(\boldsymbol{w})||_{\tilde{\mathcal{B}}}$ compatible with the following semi-inner product:

$$\langle \Phi(.)(\boldsymbol{w}), \Phi(.)(\boldsymbol{w}') \rangle_{\mathcal{B}} = \langle \boldsymbol{w}, \boldsymbol{w}' \rangle_{\mathcal{F}} \tag{13}$$

is a $\mathcal{U}$-valued RKBS with reproducing kernel $\mathcal{K}$ given in (11).

*Proof.* We show our result in the case J=1 and can be directly extended to any cardinality J. By hypothesis, $\mathcal{V}$ and $\mathcal{U}$ are a uniform Banach space and so is $\mathcal{L}(\mathcal{V},\mathcal{U})$. We hence define the feature map $\Phi$ as defined by equations (11) and (12) $\Phi$ and noting here that $\mathcal{F} = \mathcal{L}(\mathcal{V},\mathcal{U})$:

$$\Phi : \mathcal{V} \mapsto \mathcal{L}(\mathcal{L}(\mathcal{V},\mathcal{U}),\mathcal{U}) \tag{14}$$

$$\boldsymbol{v} \mapsto \Phi(\boldsymbol{v}) = \left( \boldsymbol{l} \mapsto \boldsymbol{l}(\boldsymbol{v}) \right) \tag{15}$$

In particular, by considering the uniform space $\tilde{\mathcal{F}} = \{ \boldsymbol{l} \in \mathcal{L}(\mathcal{V},\mathcal{U}) \mid \exists\ \boldsymbol{v}' \in \mathcal{V}, \boldsymbol{u} \in \mathcal{U}\ \ \boldsymbol{l} = W_{\boldsymbol{\theta}}(A_{\boldsymbol{\theta}}^1(.,\boldsymbol{v}').V_{\boldsymbol{\theta}}(\boldsymbol{u})) \} \subset \mathcal{F}$, we have the following relation:

$$\forall (\boldsymbol{v},\boldsymbol{l}) \in \mathcal{V} \times \tilde{\mathcal{F}}, \ \ \exists\ \boldsymbol{v}' \in \mathcal{V}, \boldsymbol{u} \in \mathcal{U}\ \ \text{s.t}\ \ \Phi(\boldsymbol{v})(\boldsymbol{l}) = W_{\boldsymbol{\theta}}(A_{\boldsymbol{\theta}}^1(\boldsymbol{v},\boldsymbol{v}').V_{\boldsymbol{\theta}}(\boldsymbol{u})), \tag{16}$$

identifying the adjoint $\Phi^\dagger(\boldsymbol{v}) : \mathcal{U} \mapsto \mathcal{L}(\mathcal{V},\mathcal{U})$ as $\Phi^\dagger(\boldsymbol{v}) : \boldsymbol{u} \mapsto \left( \boldsymbol{v}' \mapsto W_{\boldsymbol{\theta}}(A_{\boldsymbol{\theta}}^1(\boldsymbol{v},\boldsymbol{v}').V_{\boldsymbol{\theta}}(\boldsymbol{u})) \right)$ and verifying the kernel relation:

$$\forall (\boldsymbol{v},\boldsymbol{v}') \in \mathcal{V}^2, \ \ \Phi(\boldsymbol{v}')(\Phi^\dagger(\boldsymbol{v})) = \mathcal{K}(\boldsymbol{v},\boldsymbol{v}') = W_{\boldsymbol{\theta}}(A_{\boldsymbol{\theta}}^1(\boldsymbol{v},\boldsymbol{v}').V_{\boldsymbol{\theta}}(.)) \tag{17}$$

Furthermore, by bijectivity of the duality map on $\tilde{\mathcal{F}} \subset \mathcal{L}(\mathcal{V},\mathcal{U})$ that $\overline{span}\{(\Phi^\dagger(\boldsymbol{v})(\boldsymbol{u}))^*, \boldsymbol{v} \in \mathcal{V}, \boldsymbol{u} \in \mathcal{U}\} = \tilde{\mathcal{F}}^*$. The application of the feature map characterization of $\mathcal{K}$ on $\tilde{\mathcal{F}}$ allows to conclude.

$\square$

## S.2 Numerical implementation

### S.2.1 Loss functions and evaluations

**Definition of loss function** ⋄ In the case of operator regression, we meta-train models with respect to the Mean-Squarred error (MSE) over $I$ test pairs $(\boldsymbol{v}_i, \boldsymbol{u}_i)_{i \leqslant I}$ of the meta-train set and $K$ evaluation points $(x_k)_{k \leqslant K}$ of the domain of the output functions in $\mathcal{V}$:

$$\mathcal{L}(\tilde{\mathcal{O}}, \mathcal{D}_{\mathcal{O}}) = \frac{1}{I} \sum_{i \leqslant I} \tilde{\mathcal{L}}(\tilde{\mathcal{O}}(\boldsymbol{v}_i), \boldsymbol{u}_i) = \frac{1}{I.K} \sum_{i \in} \sum_{k \leqslant K} ||\tilde{\mathcal{O}}(\boldsymbol{v}_i)(x_k) - \boldsymbol{u}_i(x_k)||_2^2 \tag{18}$$

In the case of experiment 1 (ADR equation), $(x_k)_{k \leqslant K}$ corresponds to equally spaced points $(x_k)_{k \in [\![0,100]\!]}$ on the domain $[0,1]$. For experiment 2 (2D Burgers equation), $(x_k)_{k \leqslant K}$ corresponds to uniform 2D mesh $(x_{k,p})_{k \in [\![0,64]\!], p \in [\![0,64]\!]}$ discretizing the domain $[0,1] \times [0,1]$. For experiment 3 (Climate modeling), as stated in the main text, $(x_k)_{k \leqslant K}$ corresponds to 2D mesh $(x_{k,p})_{k \in [\![0,720]\!], p \in [\![0,720]\!]]}$ spanning the domain $[0, 180°] \times [0, 360°]$. Finally for the final finite-dimensional experiment (MNIST-like datasets), evaluation points $(k)_{k \in [\![0,10]\!]}$ corresponds to indices of 10-dimensional vectors of one-hot class encodings such that $\mathcal{L}$ corresponds to:

$$\mathcal{L}(\tilde{\mathcal{O}}, \mathcal{D}_{\mathcal{O}}) = \frac{1}{I.K} \sum_{i \in} \sum_{k \leqslant 10} |\tilde{\mathcal{O}}(\boldsymbol{v}_i)(k) - \boldsymbol{u}_i(k)|^2 \tag{19}$$

**Definition of RMSE** ⋄ Similarly, in the case of operator regression, we report average Relative Mean-Squarred Errors (RMSEs) defined as:

$$\text{RMSE}(\tilde{\mathcal{O}}, \mathcal{D}_{\mathcal{O}}) = \frac{1}{I.K} \sum_{i \in} \sum_{k \leqslant K} \frac{||\tilde{\mathcal{O}}(\boldsymbol{v}_i)(x_k) - \boldsymbol{u}_i(x_k)||_2^2}{||\boldsymbol{u}_i(x_k)||_2^2} \tag{20}$$

Note that for meta-training and meta-evaluation, MSEs and RMSEs are further averaged over batches of $J'$ elements $(\mathcal{O}_j)_{j \in J'}$.

### S.2.2 Discussion on multi-head reproducing kernels

**Kernel definition** ⋄ In coherence with Wright and Gonzalez (2021), we show that different expressions of the kernel $\boldsymbol{\kappa_\theta}$ can be proposed. Specifically, we tested three expressions:

- Exp. dot product: $A_{\boldsymbol{\theta}}(\boldsymbol{v}, \boldsymbol{v}') = \exp\left(\frac{K_{\boldsymbol{\theta}}(\boldsymbol{v}))^T (Q_{\boldsymbol{\theta}}(\boldsymbol{v}'))}{\tau}\right)$
- RBF: $A_{\boldsymbol{\theta}}(\boldsymbol{v}, \boldsymbol{v}') = \exp\left(\frac{||K_{\boldsymbol{\theta}}(\boldsymbol{v}) - Q_{\boldsymbol{\theta}}(\boldsymbol{v}')||_2^2}{\tau}\right)$
- $\ell_2$-norm: $A_{\boldsymbol{\theta}}(\boldsymbol{v}, \boldsymbol{v}') = ||K_{\boldsymbol{\theta}}(\boldsymbol{v}) - Q_{\boldsymbol{\theta}}(\boldsymbol{v}')||_2^2$

Note that for each kernel expression, we still perform a normalization operation $\boldsymbol{v} \mapsto \frac{A_{\boldsymbol{\theta}}(\boldsymbol{v}, \boldsymbol{v}_i)}{\sum_{i \leqslant I} A_{\boldsymbol{\theta}}(\boldsymbol{v}, \boldsymbol{v}_i)}$ over the entire set $(\boldsymbol{v}_i)_{i \leqslant I}$ without loss of generality. We report below regression RMSE for the ADR experiment with the different expressions for the linear function $A_{\boldsymbol{\theta}}(\boldsymbol{v}, \boldsymbol{v}')$ for different dataset sizes. The two first expressions yield similar result in the ADR experiment at an equal compute cost. For coherence, we present all other results with the "exponentiated dot product" kernel definition.

| KERNEL EXPRESSION | s=10 | s=100 | s=500 |
|---|---|---|---|
| EXP. DOT PRODUCT | $2.71e-3$ | $2.39e-4$ | $1.79e-4$ |
| RBF | $8.71e-3$ | $3.46e-4$ | $3.22e-4$ |
| $\ell_2$-NORM | $1.71e-2$ | $6.98e-4$ | $7.33e5$ |

**Table S.1:** Results from variation of the *Transducer* kernel constructions in the ADR experiment. Note that contrary to other definitions, the $\ell_2$-based kernel does not generalize to dataset cardinalities beyond those seen in the meta-training set.

### S.2.3 Details on model hyperparameters and architecture

**Discretization** ⋄ As mentionned in the main text, in order to manipulate functional data, our model can accomodate previous forms of discretization. We particularly tested two different forms of discretization discussed in (Li et al., 2020) and (Lu et al., 2019).

- In most of our experiments, we apply the *Transducer* model after performing a Fast Fourier transforms (FFT) of the considered input and output functions, and transform the *Transducer*'s output back to form estimates at arbitrary resolution. More specifically, we apply our model on the $d$-dimensional finite vector formed by the first modes of the Fourier transform, and discard the rest of the function spectrum. For experiments with 2D fields, we describe more precisely in section S.3.2 how we combine the 2D FFT with our model.

- We also tried a 'branch' and 'trunk' networks formulation of the model as in DeepONet (Lu et al., 2019). Specifically, the branch network $g : \mathcal{V} \mapsto K^P$ correspond to the *Transducer* network which outputs the weight parameters $(k_p)_{p \leqslant P}$ for the functional basis learned by the 'trunk' networks $f : D \mapsto K^P$ where $D$ corresponds to the domain of $\mathcal{U}$. Hence, the transducer model reads:

$$\forall \boldsymbol{x} \in D \;\; \mathcal{T}(\mathcal{D}_{\mathcal{O}})(\boldsymbol{v})(\boldsymbol{x}) = \sum_{p \leqslant P} g_p(\mathcal{D}_{\mathcal{O}})(\boldsymbol{v}).f_p(\boldsymbol{x}) \qquad (21)$$

We tested this approach in the ADR experiment by directly feeding the functions values $(\boldsymbol{v}_i(x_k))_{k \leqslant 100}$ and $(\boldsymbol{u}_i(x_k))_{k \leqslant 100}$ of the uniformly discretized domain of $\mathcal{V}$ and $\mathcal{U}$. We noted that performance was slightly worse than the Fourier method as we did not perform additional tuning such as feature augmentation for the branch network. For coherence, we kept the Fourier transform for the other experiments.

**Feedfoward networks definition** ⋄ For $F_\theta^\ell$ and $G_\theta^\ell$, we use a simple feedfoward network architecture defined as Layer normalization (Ba et al., 2016) followed by one layer perceptron with GeLU activation and did not performed architectural search on this part of the network.

**Architecture hyperparamters** ⋄ We present in the following table the particular architectural choices for each experiment.

| EXPERIMENT | DEPTH | MLP DIM | DIM $d$ | #HEADS | DIM HEADS |
|------------|-------|---------|---------|--------|-----------|
| ADR | 1-16 | 100 | 50 | 32 | 16 |
| BURGERS | 10 | 800 | 800 | 64 | 16 |
| CLIMATE | 6 | 512 | 512 | 40 | 16 |
| MNIST | 2 | 256 | 784 | 32 | 32 |

**Table S.2:** Summary of the architectural hyperparameters used to build the *Transducer* in the four experiments. 'Depth' corresponds to network number of layers, 'MLP dim' to the dimensionality of the hidden layer representation in $F_\theta^\ell$ and $G_\theta^\ell$, $d$ to the dimension of the discrete function representations.

### S.2.4 Details on meta-training

As stated, we used for all experiments, the same meta-training procedure. We optimized *Transducer* models using the Adam optimizer (Kingma and Ba, 2014) for a fixed number of epochs with learning rates halved multiple times across meta-training.

| EXPERIMENT | # OF EPOCHS | LEARNING RATE | DIM HEADS |
|------------|-------------|---------------|-----------|
| ADR | 200 | $1e-4$ | 50 |
| BURGERS | 200 | $1e-4$ | 800 |
| CLIMATE | 200 | $1e-4$ | 512 |
| MNIST | 500 | $1e-4$ | 784 |

**Table S.3:** Summary of the meta-learning hyperparameters used to meta-train the *Transducer* in our four experiments.

## S.3 Experiments

In this section, we provide additional details with respect to data generation and model evaluation for each experiments discussed in section (5) of the main text.

### S.3.1 Advection-Diffusion-Reaction operators

**Data generation** – For our experiment, we collect a meta-dataset of $N = 500$ datasets of the advection-diffusion-reaction trajectories on the domain $\Omega = [0,1] \times [0,1]$ by integrating the following equations:

$$\forall n \in [\![1, 500]\!], \quad \partial_t \boldsymbol{s}(x,t) = \underbrace{\nabla \cdot (\boldsymbol{\delta}_n(x) \nabla_x \boldsymbol{s}(x,t))}_{\text{diffusion}} + \underbrace{\boldsymbol{\nu}_n(x) \nabla_x \boldsymbol{s}(x,t)}_{\text{advection}} + \underbrace{\boldsymbol{k}_n \cdot (\boldsymbol{s}(x,t))^2}_{\text{reaction}} \quad (22)$$

We use an explicit forward Euler method with step-size $1e^{-2}$, storing all intermediate solutions on a spatial mesh of 100 equally spaced points. Hence, our discretized reference trajectories are of dimensions $100 \times 100$. For each operator $\mathcal{O}_n$ we generate spatially varying diffusion and advection coefficients as random function $\boldsymbol{\delta}_n(x) : [0,1] \mapsto \mathbb{R}$ and $\boldsymbol{\nu}_n(x) : [0,1] \mapsto \mathbb{R}$ as well as a random scalar reaction coefficient $\boldsymbol{k}_n$. Defining $\mathcal{G}(0, k_l(x_1, x_2))$ the one-dimensional zero-mean Gaussian random field with the covariance kernel:

$$k_l(x_1, x_2) = e^{\frac{-\|x_1 - x_2\|^2}{2l^2}} \quad (23)$$

and lenght-scale parameter $l = 0.2$, as well as a boundary mask function $m : [0,1] \mapsto [0,1], m(x) = 1 - (2x - 1)^{10}$ (to comply with Dirichlet boundary condition and preserve numerical computation stability), we sample $\boldsymbol{\delta}_n(x)$ and $\boldsymbol{\nu}_n(x)$ according to the following equations:

- **diffusion** $\boldsymbol{\delta}_n(x) = 0.01 \times u_n(x)^2 \times m(x)$ where $u_n \sim \mathcal{G}(0, k_{0.2}(x1, x2))$
- **advection** $\boldsymbol{\nu}_n(x) = 0.05 \times y_n(x) \times m(x)$ where $y_n \sim \mathcal{G}(0, k_{0.2}(x1, x2))$
- **reaction** $\boldsymbol{k}_n \sim \mathcal{U}([0, 0.3])$.

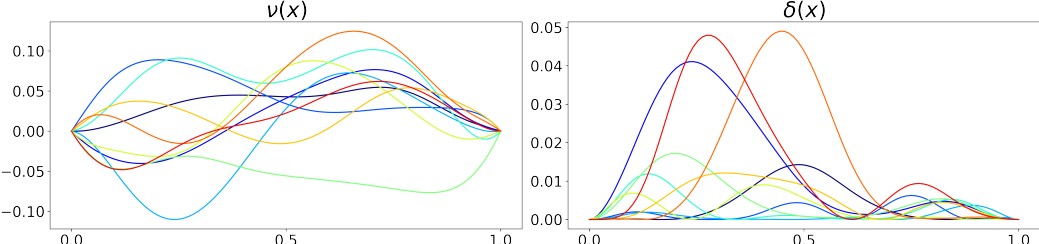

**Figure S.1:** Examples of sampled functions $\boldsymbol{\delta}(x)$ and $\boldsymbol{\nu}(x)$ used to build operators $\mathcal{O}_n$.

Furthermore, we collect for each dataset $i = 100$ trajectories with each different initial state $s(x,0) = \boldsymbol{v}_i(x)$, where functions $\boldsymbol{v}_i(x)$ are sampled according to the following:

- **initial state** $\boldsymbol{v}_i(x) = m(x) \times u_i(x)$ where $u_i \sim \mathcal{G}(0, k_{0.2}(x1, x2))$.

For meta-testing, we sample $N = 500$ new datasets of the same generic advection-diffusion-reaction equation with new parameters $\boldsymbol{\delta}_n(x), \boldsymbol{\nu}_n(x), \boldsymbol{k}_n(x)$, for up to 1000 different initial states $\boldsymbol{v}_i(x)$. We present below example of function profiles present in the meta-datasets.

**Training** ⋄ We train Tranducers for 200K gradient steps. At each training step, we randomly draw a single operator $\mathcal{O}_n$ from the meta-training set and isolate the pairs $(\boldsymbol{v}_i, \boldsymbol{u}_i)_{i \leqslant I} = (s_i(x,0), s_i(x,1))_{i \leqslant I}$ to form the set $\mathcal{E}_{\mathcal{O}_n}$. We sample a "query" subset $\mathcal{Q}$ of $J = 10$ pairs from $\mathcal{E}_{\mathcal{O}_n}$ to be regressed and form the input to our model by concatenating pairs of the query set $\mathcal{Q}$ (with output elements $(\boldsymbol{u}_i)_{i \in \mathcal{Q}}$ set to zero), with a non-overlapping set of $I \in [\![20, 100]\!]$ example elements drawn from $(\boldsymbol{v}_i, \boldsymbol{u}_i)_{i \notin \mathcal{Q}}$. We train our model to minimize the sum of $L_2$ error between each output function of the set $Q$ and its corresponding ground truth $\boldsymbol{u}(x) = \mathcal{O}_n(\boldsymbol{v})(x) = \boldsymbol{s}(x,1)$ at the 100 discretized positions.

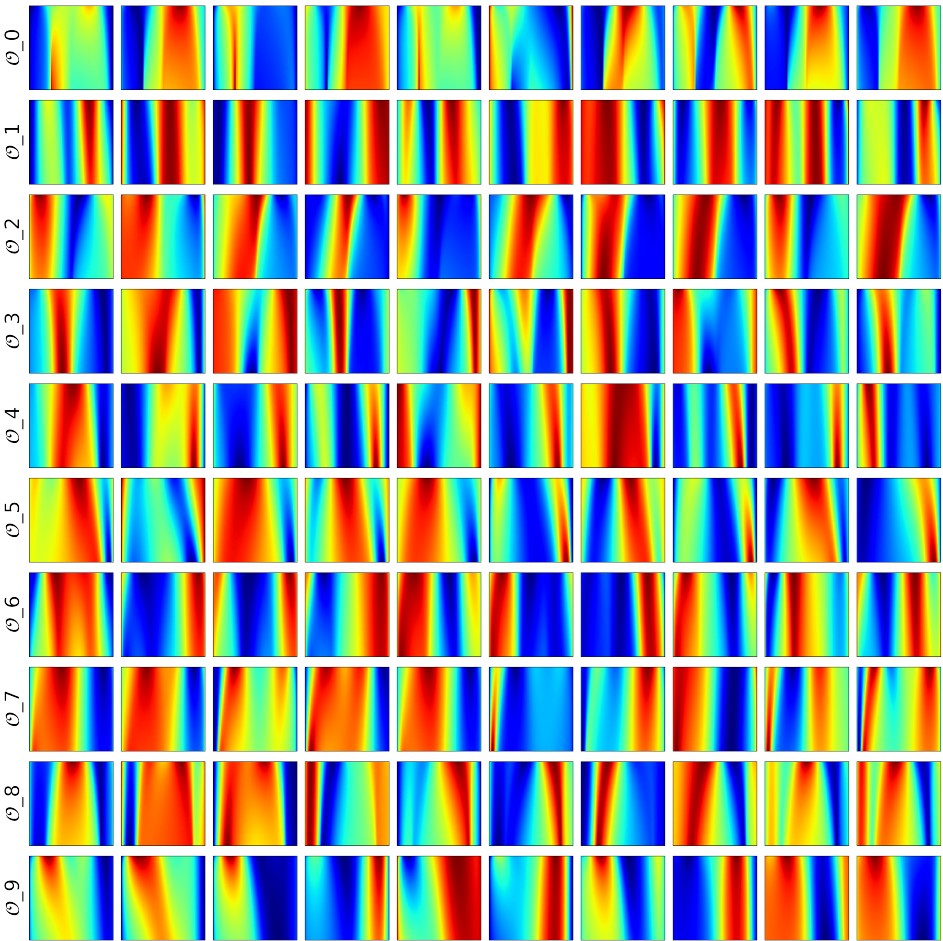

**Figure S.2:** Examples of advection-diffusion-reaction datasets (different operators by row) present in the meta-test set.

**Baselines** – In order to implement the baseline regression algorithms, we use the scikit-learn library (Pedregosa et al., 2011) for decisions trees, $K$-nearest neighbours and Ridge regression. We specifically tuned Ridge regression using cross-validation and selected the best-performing 'RBF' kernel with regularisation $lambda = 1e^{-3}$. For FNO (Li et al., 2020), we use the official PyTorch implementation provided by authors and defined for each regression, a 4-layer deep 1-dimensional FNO network with 16 modes and 64-dimensional $1 \times 1$ convolutions. For DeepOnet (Lu et al., 2019), we implement our own PyTorch version with 4 hidden layers of 50 hidden units with ReLU activation for the branch and trunk networks. For the FNO-MAML version, we kept the same model definition while testing several adaptation budget is 10/50/100 gradient steps with learning rate $\lambda = 1e - 2/7e - 3/5e - 3$ respectively.

**Extrapolation experiment** – In this task, we modify the generative process of the considered operators by changing the lenght-scale parameter $l$ used to produce functions $\boldsymbol{\delta}(x)$ and $\boldsymbol{\nu}(x)$, as well as the target time $t$ used to define the operator output.

### S.3.2 Burger's equation

**Generation** ⋄ In order to produce the meta-datasets of our second experiment, we use the ΦFlow library (Holl et al., 2020) that allows for batched and differentiable simulations of fluid dynamics and available at `https://github.com/tum-pbs/PhiFlow`. Following the same methodology as experiment 1, we generate batches of the state evolution of random functions $(\boldsymbol{v}_i) : \mathbb{R}^2 \mapsto \mathbb{R}^2$ defined

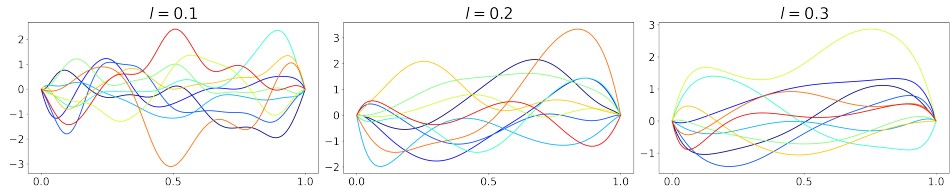

**Figure S.3:** Examples of the spatial function sampled with carying lenght scale parameter $l \in [0.1, 0.2, 0.3]$

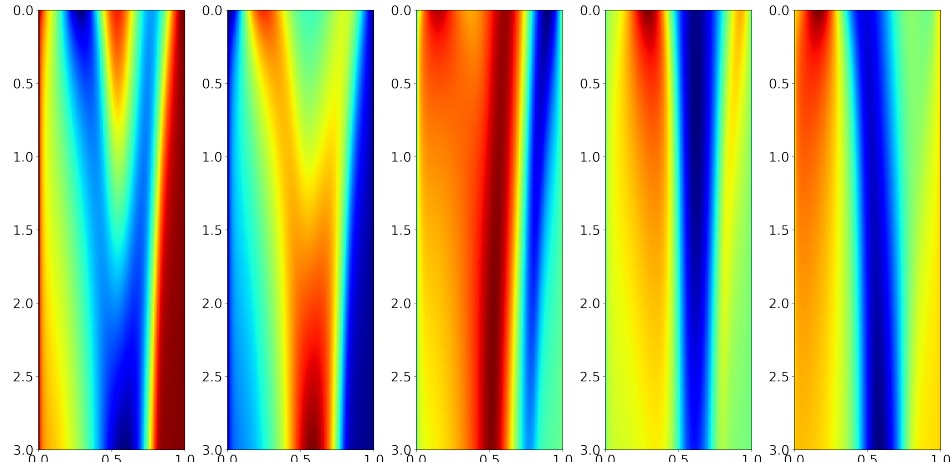

**Figure S.4:** Examples of ADR state evolution forming a set of operators with the same generative parameters but time $t$ allowed to vary in $[0, 3]$

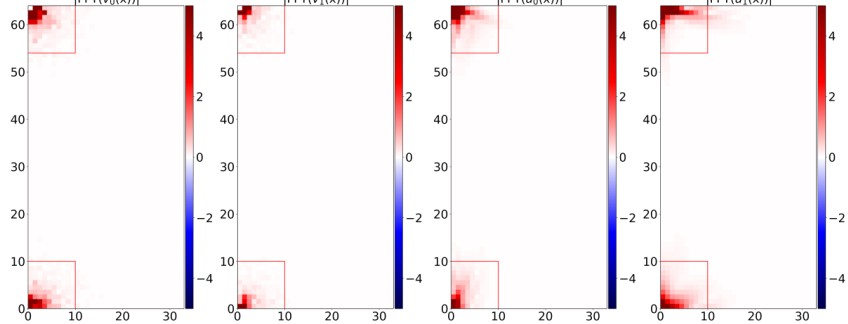

**Figure S.5:** Magnitude of the complex coefficients of the Fourier transform of an exemple pair of input and output functions $(\boldsymbol{v}(\vec{x}), \boldsymbol{u}(\vec{x}))$ in the two coordinates dimension. For every pair, the majority of the signal lies in the two the red quadrants.

on the domain $\Omega = [0, 1]^2$ at a resolution of $64 \times 64$ through different parametrization of equation (**??**). We form a meta training set of 200 operator datasets for different parameters $\boldsymbol{\nu} \in [0.1, 0.5]$ each of cardinality $I = 100$, and meta testing set of 200 different operator datasets with the same cardinality. Here, we consider vector fields input functions $\boldsymbol{v}(\vec{x})$ whose coordinates $(\boldsymbol{v}_1(\vec{x}), \boldsymbol{v}_2(\vec{x}))$ are drawn each from a two-dimensional zero-mean Gaussian random fields with uniform exponential covariance function and correlation length $l = 0.125$.

**Discrete Fourier representation** – Since we are dealing with high-dimensional inputs, we perform kernel regression on the 2D fast Fourier transforms of our model. To reduce further dimensionality, since the FFT of a real signals is Hermitian-symmetric, we pass as input to our model only the flattened $10 \times 10$ upper and lower quadrants of the Fourier transform coefficients, since we verified that those are sufficient to reconstruct the signal at relative error level of $1e-5$. (We present examples

of the 2D FFT of our signal.) After regression, we reconstruct our model estimate in the spatial domain at the desired $64 \times 64$ resolution and train for the $L_2$ distance against ground truth.

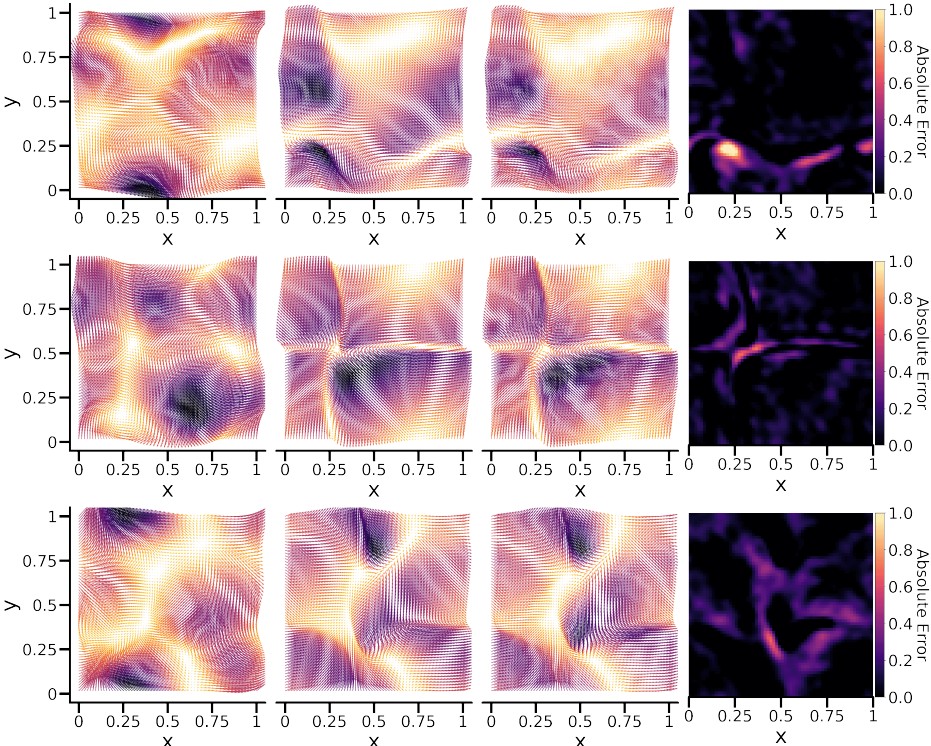

**Figure S.6:** Illustrative examples of initial ($t = 0$), target ($t = 10$) and *Transducer* estimation of the vector field $s(\vec{x}, t)$ discretized at resolution $64 \times 64$ over the domain $[0, 1]^2$ for the Burger's equation experiment. The last panel represents absolute error compared to ground truth.

**Spectral clustering** ⋄ As a baseline for the outlier detection experiment, we used the spectral clustering algorithm (Yu and Shi, 2003) implemented in the `Scikit-learn` on the same FFT preprocessing transformation of the output elements $(u_i)_{i \leqslant I}$ that is discussed above and specifying the number of clusters $C = 2$. We tried to tune the clustering algorithm in the embedding space either using K-means or a kernel formulation. The tested variations yielded no significant difference in performance.

### S.3.3   Climate modeling

**ViT modification** ⋄ In order to tackle the high-resolution climate modeling experiment, we take inspiration from Pathak et al. (2022), which combines neural operators with the patch splitting method of Vision Transformer (ViT) (Dosovitskiy et al., 2021). Specifically, we split input and output functions into patches of size $40 \times 40$. Since both models operations preserves dimensionality, we interleaves *Transducer* layers that apply kernel transformations $\kappa_\theta$ along the batch dimension with ViT layers performing spatial attention on the set of patched output function representations $(u_i)$. We drop positional encoding but reduce spatial attention to the neighboring patches for each patch position through masking. We compare this bi-attentional model to a vanilla ViT model that learns by induction a single map from temperature $\mathcal{V}$ to pressure $\mathcal{U}$. We double the depth of this baseline to $L = 12$, in order to match number of trainable parameters.

**Data** ⋄ We take our data from ERA5 reanalysis (Hersbach et al., 2020), that is freely available on the Copernicus https://cds.climate.copernicus.eu/cdsapp#!/dataset/reanalysis-era5-land?tab=overview. Surface and temperature pressure are re-gridded from a Gaussian grid to a regular Euclidean grid using the standard interpolation scheme provided by the Copernicus Climate Data Store (CDS) to form 2D fields that we further interpolate in the longitude

dimension to obtain images of size $720 \times 720$. Although the ERA5 possess hourly estimates, we subsample the dataset by considering only measurement at 12:00am UTC every day.

**Training** ⋄ As mentioned in the main text, we trained our model to predict variables for 5 days randomly sampled from a 20-day window and condition the *Transducer* with remaining 15 days. We do not explore larger settings due to GPU memory constraints.

### S.3.4 MNIST-like dataset classification

**Training** ⋄ We report results from Kirsch et al. (2022) for baselines and train and evaluate our model on datasets versions provided by the `torchvision` library. For this version, we directly treat the images inputs $(v_i)_i$ as 784-dimensional vectors and the outputs $(u_i)_i$ as 10-dimensional vectors. We do not perform intermediary non-linear transformations $G_\theta^\ell$ for the outputs representations. We haven't performed extensive hyper-parameter search for this experiment in terms of learning rate, head dimensions or kernel expression but simply noted that a deeper 4-layer version of the model was giving similar performance results.