# OpenReview forum: "Learning Functional Transduction"
_NeurIPS.cc/2023/Conference — NeurIPS 2023 spotlight_

### Official Review · Reviewer_w6wT · 2023-06-19

**Soundness:** 3 good
**Presentation:** 3 good
**Contribution:** 2 fair
**Rating:** 5
**Confidence:** 2

**Summary:**

This paper proposes a new deep learning approach for the problem of meta-learning. Inspired by the theory of reproducing kernel Banach space, the proposed method jointly trains a deep transformation as a representation, and a parametrized kernel function $K(vi, \cdot)$ of the problem instances, at the meta-training stage. Experimental results validate the effectiveness of the proposed method.

**Strengths:**

The motivation is strong, as the problem of meta-learning is a rather important problem in the community. The link between meta-learning and the theory of RKBS discussed in this paper may bring some new ideas.

**Weaknesses:**

My main concern comes from the fact that the proposed method also needs a meta-training procedure. At this point, the comparison in some experiments (for example, section 5.1, if there is no mis-understanding) seems somewhat unfair. Also, it would be better if there is some more comparison between the proposed method and some other meta-learning approaches.

**Questions:**

It would be better if the authors could give a thorough discussion about their method’s superiority, as there are many meta-learning approaches that are in a similar style, that is, first training a meta model in the dataset containing sufficient information of the task space, then apply some task-specific procedures in the target task. It would be better if the authors could clearly state several specific points, at which their method is better than most of the others in the literature.

**Limitations:**

No.

---

> ### Author Rebuttal · Authors · 2023-08-08
>
> Thank you very much for your time and interest in reviewing our work. We would like to bring some answers and clarifications in relation to your comments, and hope they might decide you to increase your score. It seems that your concern is about not properly evaluating our approach against other meta-learning approaches, so we try to emphasize this aspect in the specific replies below.
>
>  > Regarding _“[…] the proposed method jointly trains a deep transformation as a representation, and a kernel function […] of the problem instances”_.
>  - We want to clarify that no vectorial representation is carried over from the meta-training procedure. Rather, the meta-optimization fit the kernel function $K: V \times V \mapsto U$ which is expressed as an iterative application of the intermediate parametric function $k_{\theta}^{l}$ (equation 5). This kernel function is then directly applied to each new dataset along with query points to produce new output estimates.
>
>  > Regarding _“the comparison in some experiments […] seems somewhat unfair.”_
>  - We break down our reply in different arguments: First, we dedicated a whole experiment on a task previously used to benchmark meta-learning models in [1], namely MNIST-like datasets classification and compared against several state-of-the-art meta-learning systems, which directly provides evidence that the Transducer meta-learned adaptation transfer better to new tasks when meta-trained on the **same task distribution**. Second, in section 5.3, we show that our formalism can enhance the performance a regular supervised method on a large natural dataset, even though, it is again trained with the **exact same amount of data**. Third, in section 5.1 we emphasize that our original meta-learning approach can find better predictive solutions than previous neural operator approaches even when neural operators have access to large regimes of data (~1000 training trajectories). This is itself an interesting result as meta-learning systems have been very scarcely applied to the problem operator regression. To our knowledge, there is little literature on applying MAML or RNN-adaptation to neural operator. However, to convince you more of the interest of our approach, we are working on complementing section 5.1 with a comparison with a MAML-like adaptation procedure applied to FNO/DeepONet and will communicate the results as soon as possible. (Note however, that such adaptation technique is much slower than our feedforward adaptation process, see "Speed" below)
>
>  > Regarding _“It would be better if the authors could give a thorough discussion about their method’s superiority […]”_
>  - In the experimental section, we tried to provide different arguments showcasing the benefits of our method in several domains, against different class of models and along different features. We synthesize the main aspects below:
>    - **Accuracy**: First, our original transductive system can fit solution of operator regression problems at the level of accuracy of previous (gradient-based) neural operator regression approaches. (Section 5.1). Conversely, we show that our system is also a better meta-learner than several state-of-the-art meta-learning approaches on a task where they have been extensively benchmarked. (Section 5.4). Finally, we show that our philosophy can boost purely inductive systems in section 5.3.
>    - **Speed**: The adaptation mechanism of our model is parallel by nature and bypass the need for sequential adaptation such as in gradient-or-RNN-based adaptation. This in turn translate to significant gain in computation time, showcased in section 5.2 (outliers detection), where 5000 operator regression instances can be performed in under a few seconds while other meta-learning systems would need a much greater time because of their sequential adaptation process (I.e iterations of gradient descent for instance). This can potentially unlock new operator regression applications requiring fast sampling of multiple fitted models (such as bagging, conformal prediction or Monte-Carlo sampling in the operator space…). We will add a specific remark regarding this point in section 5.2.
>    - **Theoretical soundness and interpretability**: As you noted, our model builds on clear theoretical results regarding the existence and analytical form of the solutions to the considered regression problems. This is in contrast to few-shot gradient-based learning, which performs only a few gradient step (while paradoxically gradient descent is an asymptotic process by nature) and offer, in the general setting, weak guarantees regarding the quality of the found solutions. Furthermore, since our model takes explicitly as inputs the whole dataset, it offers a direct possibility, contrary to other approaches, to apply sensitivity analysis of the model with respect to each exemple for learning each task regression instance, which can be leverage to directly analyse to interpret model decisions.
>    - **Originality**: The original use of the theory of RKBS which generalise that of RKHS allows to encompass both operator (infinite-dimensional) and regular finite-dimensional regression problems in a single meta-regression framework and is, as such, worthy of exploration and analysis, along gradient-based or RNN-based adaptive systems, as it might open new discussion and connections with existing questions such as the interpretation in terms of kernel of in-context learning in attentional models.
>
> We thank you again for your time and remain committed to answer any additional clarification that you might need and engage with you further during the discussion period.

---

> > ### Comment · Reviewer_w6wT · 2023-08-12
> >
> > Thank you for your patience to resolve my confusions! I have now understood the experimental settings in section 5. Though I keep my point that the work has sound theoretical grounding(mostly due to the lack of understandings of DNNs), I admit that the idea of meta-learning neural operators has its novelty. I will no longer stand in the way of acceptance.

---

> > > ### Author Response · Authors · 2023-08-19
> > > **Second reply to reviewer w6wT**
> > >
> > > To reviewer w6wT,
> > >
> > > We thank you for your positive re-evaluation of our work. Moreover, we are happy that you praise the theoretical aspect of our work in your reply. To help definitely convince you, as we proposed in our rebuttal, we would like to complement our experimental results in section 5.1 (ADR equations) by providing a comparison with model-agnostic meta-learning (MAML) applied to FNO with different inner training budgets (10/50/100 gradient steps with inner learning rate at 1e-2/7e-3/5e-3), in line with you main suggestion (as well as that of reviewer Sczv). We synthesize the results in terms of RMSE and fine-tuning time in the following table:
> > >
> > > | RMSE / Adaptation time (sec.) | 10 gradient steps  | 50 gradient steps  |  100 gradient steps
> > > |---|---|---|---|
> > > | FNO-MAML  | $6.5e^{-1} // 2.6e^{-1}$  | $3.1e^{-1} // 6.7e^{-1}$ | $1.4e^{-1}//2.1e^{0}$  |
> > >
> > > As you can see, on this task, a meta-learned parameter initialization with a limited fine-tuning budget does not improve over the previously tested fine-tuned approaches. More importantly, this approach is much less accurate and computationally more intense than our system (due to the need for sequential gradient computation). We hope that this control experiment will help dissipate your concern and contrastively demonstrate the potential value of our own approach. We remain happy to answer to any additional comment before the end of the discussion period.

---

### Official Review · Reviewer_f2BS · 2023-06-27

**Soundness:** 4 excellent
**Presentation:** 4 excellent
**Contribution:** 3 good
**Rating:** 7
**Confidence:** 3

**Summary:**

This paper proposes a method for transductive learning based on reproducing kernel banach spaces (RKBS). The resulting model is capable of learning *in-context* in the sense that given a new instance of a learning problem or a new dataset $\mathcal{D}$, it can infer the resulting functional relationship at inference time. The authors show that transformer attention layers can be considered reproducing kernels for this model. They provide a meta training objective that allows them to learn the kernel operations $\kappa^\ell$ and nonlinearities $F^\ell$ which generalize well over the distribution $\mathfrak{D}$ of possible datasets. Once trained, the model can generate functions for new datasets.

The authors test the proposed ideas to learn operators $\mathcal{O}$ for PDEs including advection reaction diffusion equation and Burgers' equation as well as climate modeling. Lastly, they test on a MNIST like task where there are pixel permuted and class permuted versions of the dataset. In all of these settings they identify benefits to their approach over several existing benchmarks.

**Strengths:**

The paper provides a nice framework and perspective to think about transductive learning through reproducing kernels. To the best of my knowledge, this is a novel idea.

The paper also provides several experiments that showcase the benefits of the approach, especially in PDE modeling.

**Weaknesses:**

As one of the primary motivations of the model is a meta-learning algorithm capable of in-context learning, I think that an experiment involving in-context learning of language patterns would greatly strengthen the paper. Most of the experiments at this point are for PDEs but the proposal is much broader.



**Questions:**

On the MNIST experiment, how well would a pure supervised algorithm (perhaps with a provided context signal to indicate permutation) trained on all instances of the data perform? I am wondering if the meta-learning objective outperforms standard supervised learning because it sees a larger amount of total data.

**Limitations:**

The authors addressed their need to work with finite dimensional outputs for the PDE modeling (Fourier coefficients) and also acknowledged the computational and statistical requirements to optimize the metal learning objective.

---

> ### Author Rebuttal · Authors · 2023-08-08
>
> Thank you for your review and the interesting suggestions that you shared. We are happy about your positive assessment and we reply specifically to your comments below.
>
>  > Regarding adding "an experiment involving in-context learning of language patterns"
>  - This is an intriguing suggestion. We agree that an experiment involving natural language could definitely strenghten the interpretation of in-context learning in attentional layers as a form of reproducing kernel fitting since ICL has been predominently demonstrated  in this domain. However, analyzing natural language in terms of functional reproducing banach space of is non trivial and we chose to select more straightforward exemple tasks to illustrate our presentation. We plan on developing further this idea in a follow-up work.
>
> > Regarding the MNIST experiment.
>  - We want to emphasize that all baselines are meta-learning systems that receive the same training curriculum as our model. Hence, the difference of performance can not be attributed to training data volume. Moreover, note that providing a context signal to indicate permutation will be counter-productive at test time, since it would provide a shortcut to circumvent meta-learning of a genuine regression program, hence, preventing transfer learning on other datasets (FashionMNIST and KMNIST). On the other hand, training a supervised baseline on a large quantity of class and pixel permutation with no permutation signal is likely to provide no learning at all. We agree however with you, that quantifying how our meta-learned kernel solution converge as a function of data volume is an important future direction for our approach.
>
> We thank you again for your time and valuable feedback.

---

> > ### Comment · Reviewer_f2BS · 2023-08-11
> > **Response to Authors**
> >
> > I appreciate the authors' answers to my questions about the possibility of further ICL experiments and data volume. Though natural language experiments would strengthen the work, it could also be left as a future application. I will maintain my support for acceptance.

---

### Official Review · Reviewer_Sczv · 2023-06-30

**Soundness:** 4 excellent
**Presentation:** 4 excellent
**Contribution:** 4 excellent
**Rating:** 8
**Confidence:** 4

**Summary:**

**SUMMARY AFTER REBUTTAL**: as described below, most of my comments were addressed and the authors made a significant job in including new results. I have increase my score during the rebuttal phase and I strongly vote for acceptance.

---

Neural operators are neural networks that are trained to approximate a function-to-function mapping. This paper proposes an algorithm to perform few-shot learning of these models, where at inference time the network is given a set of examples of the operator, and an input function as the test point.

The specific algorithm they propose is built on reproducing kernel Banach spaces, where for a given prediction, the kernel and the dictionary on which it is evaluated is computed by a recursive embedding of the few-shot examples. This is inspired by the transformer architecture. When examples are continuous, they are discretized with an FFT.

They analyze a series of classical benchmarks including modeling PDEs, showing the method to perform well.

**Strengths:**

As far as I know, the idea of performing few-shot learning of neural operators is novel. These networks are used extensively in physical modelling, so the paper can have a sizable impact there.

The paper is also relatively clear (with a few exceptions, see below), especially if one considers the complexity of some underlying ideas. The connection between their method and the transformer is also interesting, but from what I understand this is extended from Wright and Gonzalez (2021).

The experiments are varied and cover a wide range of use cases.

**Weaknesses:**

1) I have found the initial discussion on the difference between transduction and inference a bit misleading. According to Vapnik, 2006 (which they cite), almost anything which is used today in ML / deep learning is inductive. Transductive methods are only those that can make predictions on a given set of test points but *cannot* operate outside those. For example, SVM is inductive in general, except for some variants such as the transductive SVMs discussed in Collobert et al., 2006. In fact, what they are calling "transductive" is what is typically called "instance-based" in ML (kNN, SVM). Their setup is a standard few-shot setup extended to operators. This also leads to some strange sentences, such as "inductive neural learning with gradient descent is compute-intensive" referred to neural networks; SVM training is also notoriously intensive (and it can also be done via gradient descent).

2) There are some "standard" methods for performing few-shot learning in the literature (e.g., MAML, prototype networks, ...). Many of these methods start from a standard neural network and adapt it to a few-shot scenario. Architectures for performing operator learning are known, and it is not clear from reading the paper why the few-shot learning methods are not immediately extensible to this setup (e.g., why can't we do MAML on a standard neural operator?), and why we need instead to resort to a more complex formulation in terms of kernels. To clarify: I think the algorithm shown here is interesting, but it's a bit hard to motivate it by reading the paper itself.

3) The computational complexity of the method is not discussed, in particular the need to execute multiple FFTs for the few-shot examples (Section "Discretization"). This also connects to recent literature on performing FFTs in an efficient way on GPUs (e.g., FlashConv). Also for testing, if I understand correctly the authors are mostly comparing their method (which is only a forward pass) with a full training of its competitors, which is a bit unfair. No baselines for few-shot learning are tested. For example, they state that their setup is "Similar to (Pathak et al., 2022)", but they do not compare.

**Questions:**

- I think rephrasing the entire "transductive" discussion in terms of few-shot learning could significantly improve the paper. At the least, objectively incorrect sentences should be amended.

- Discussing better training and test complexity is important.

- I think the paper can improve readability by providing a practical example of (5)-(6) in the more familiar case of finite-dimensional input-output spaces.

- There has been a limited amount of works that have explored using neural networks to perform Bayesian posterior inference on-the-fly on a large family of distributions (e.g., Prior-Data Fitted Networks (PFNs)). I would be curious to see a discussion on the connection with this work. On the related work part, there is also some relevant literature on recursive (recurrent) kernel evaluation which is not mentioned.

- There is a small typo on page 3: point evalutation.

- Will the code be released?

**Limitations:**

The authors are clearly discussing the limitations of their few-shot setup. However, they are not discussing the computational complexity of the method, which I think would be the biggest limitation in scaling this up to large setups. A few sentences on this would be appreciated.

---

> ### Author Rebuttal · Authors · 2023-08-08
>
> Thank you for your time and for this thoughtful review. We are happy that you seem enthusiastic about our approach and we try to answer specifically to your remarks and questions below:
>
>   > Regarding “the initial discussion on the difference between transduction and inference”.
>  - Indeed, we definitely agree with you that current deep learning techniques rely, in a vast majority, on inductive learning principles, which constitutes a starting motivation of our work. We also definitely agree that "instance-based learning" is another valid denomination regarding how decisions rules are formed in our approach. However, we disagree on your definition of transductivity (_"Transductive methods are only those that can make predictions on a given set of test points but cannot operate outside those"_) See for instance the seminal _Learning by Transduction_ [1]. (Section 1: "This is a problem of transduction, in the sense that we are interested in the classification of a particular example rather than in a general rule for classifying future examples" and section 6 ""Transduction" is inference from particular to particular; for the problem of pattern recognition, it means that, given the classifications Yi, i = 1, . .. , l,of the l points x1, ... , Xl in the training set, we are only trying to guess the classifications of the k points xl+1, ... , Xl+k in the test set." Remark 5 of the same  reference also directly agree with your denomination of "instance-based learning" which we will mention in the introduction. Finally, we reworked the expression "inductive neural learning [...] is compute-intensive" to emphasize instead that iterative optimization procedures for tackling high-dimensional problems are compute-intensive and bottlenecked bye their sequential nature.
>
> > Regarding data regimes and comparison “with standard methods".
>  - We would like to point out that our method is not restricted to the "few-shot" data regime but can be applied to problem instances with more than thousands exemples (for instance on section 5.3, we test out model with up to 1000 exemples pairs.). However, as you noted, our model needs an original meta-optimization procedure that is unusual in the recent neural operator litterature. To your suggestion, we are working on complementing section 5.1 with a comparison with a MAML-like adaptation procedure applied to FNO/DeepONet and will communicate the results as soon as possible. Note however, that such adaptation technique will be much slower than our feedforward adaptation process.
>
> > Regarding the computational complexity of the method, scaling and the cost of FFTs.
>  -  We are actually performing a single FFT/IFFT transformation as a pre/post-processing operations, in line with recent litterature on Fourier operators [2]. This largely mitigates the cost of such operation which allowed us to scale our system to vert high-resolution climatic prediction (720x720 images). Note that in this experiment, we are not interested in obtaining state-of-the-art climatic variable prediction as in Pathak et al., 2022, but we are showing that our transductive approach can augment popular inductive models in the context of operator regression problems.
>
>  > Regarding improving readability of equations 5-6 with a finite-dimensional example.
>  - To your suggestion, we are working on integrating a cartoon of the kernel computation to the exemples depicted in figure 1 to improve readability.
>
> > Regarding related work on Bayesian posterior inference and recursive kernel iterations.
>  - Thank you for these interesting suggestions, we will definitely mention this relevant litterature to our discussion. Regarding recursive kernel evaluation, are there specific pointers that you could share for us to adapt our discussion?
>
> > There is a small typo on page 3: point evalutation."
>  - Thank you for this catch.
>
> > Will the code be released?
>  - Yes we plan to release a public repository with the code to replicate the experiments as well as pre-run notebooks to help the interested reader familiarize with our model.
>
> We thank you again for your interest and this valuable feedback which definitely helps strenghten our work.
>
> [1] Learning by transduction A. Gammerman, V. Vovk, V. Vapnik. UAI'98: Proceedings of the Fourteenth conference on Uncertainty in artificial intelligence
> [2] Transform Once: Efficient Operator Learning in Frequency Domain. Michael Poli, Stefano Massaroli, Federico Berto, Jinykoo Park, Tri Dao, Christopher Ré, Stefano Ermon. NeurIPS 2022

---

> > ### Comment · Reviewer_Sczv · 2023-08-16
> >
> > Thanks for the detailed feedback. Concerning the definition of "transduction", I understand the point but I also think that "transduction" used in this sense can be misleading, especially since there are other definitions that are more common in today's literature. However, this does not impact my evaluation in any way. I will keep my score as-is waiting for the other points. For recursive kernels, the authors might be interested in this paper coming from the kernel signal processing literature, which is a bit niche but connected: https://ieeexplore.ieee.org/document/6722955

---

> > > ### Author Response · Authors · 2023-08-19
> > > **Second reply to reviewer Sczv**
> > >
> > > To reviewer Sczv,
> > >
> > > Thank you for your reply as well as the interesting reference. We are happy that our rebuttal was informative. In addition to these, as we mentioned in our rebuttal, we would like to complement our experimental results in section 5.1 (ADR equations) by providing a comparison with an other meta-learning method. While we re-emphasize that meta-learning of neural operators is a new topic with no standard approaches, we tried to your suggestion to apply gradient-based model-agnostic meta-learning on our meta-dataset of operators to the same FNO model with different inner training budgets (10/50/100 gradient steps with fixed inner learning rate at 1e-2/7e-3/5e-3). We synthesize in the following table the results in terms of RMSE and fine-tuning time that will complement table 1 of the paper.
> > >
> > > | RMSE / Adaptation time (sec.) | 10 gradient steps  | 50 gradient steps | 100 gradient steps
> > > |---|---|---|---|
> > > | FNO-MAML | $6.5e^{-1} //  2.6e^{-1}$ | $ 3.1e^{-1} // 6.7e^{-1}$ |  $1.4e^{-1} // 2.1e^{0}$ |
> > >
> > > As you can see, on this task, a meta-learned parameter initialization with a limited fine-tuning budget does not improve over the previously tested fine-tuned approaches. More importantly, this approach is much less accurate and computationally more intense than our system (due to the need for sequential gradient computation). These complementary results further validate the interest of our own approach. We agree that this comparison will also help the reader better situate our approach in the literature.
> > >
> > > We remain happy to answer to any additional comment before the end of the discussion period.

---

> > > > ### Comment · Reviewer_Sczv · 2023-08-19
> > > >
> > > > I really appreciate the effort that the authors have put into addressing this specific concern. As I believe this is a very strong addition to the paper, I have increased my score to a Strong Accept and most other scores to Excellent.

---

### Author Rebuttal · Authors · 2023-08-09

We would like to thank again all three reviewers for their interest and precious feedback. We are happy that this work has been positively regarded by reviewers Sczv and f2BS while reviewer w6wT seemed less confident. We answered directly to specific remarks and questions in each review thread (see below) while integrating common suggestions to our text which we believe are strenghtening our proposal.

As a synthesis, all three reviewers recognised that our work constituted an original proposal with a _"strong motivation"_ and _"novel ideas"_ that can potentially impact research at the intersection of neural operators and meta-learning. Reviewer Sczv noted a _"clear"_ presentation of ideas and reviewer f2BS liked "a nice framework and perspective"_. A central concern of reviewer w6wT was _"to give a thorough discussion about [our] method's superiority" against other meta-learning approaches. However this is specifically the aim of section 5.4 which compares our system against existing meta-learning approaches, while we demonstrate several original results in the other experimental sections: Namelly, In-context learning in infinite-dimensional spaces in section 5.1, outliers detection in section 5.2 and scaling/inductive model boosting in section 5.3. We are working on complementing our results to reviewer w6wT suggestion, but we note simultaneously that reviewer Sczv underscored that our _"experiments are varied ans cover a wide range of use case"_ and that reviewer f2BS remarked that we _"identify benefits"_ in all of them.

We will be happy to discuss further these points during the discussion period and remain commited to answer any additional question that may arise after our rebuttal.

The authors

---

### Decision · Program_Chairs · 2023-09-21

**Decision:**

Accept (spotlight)

**Comment:**

The paper presents a quite interesting framework for meta-learning that uses neural networks to learn a function in an RKBS. The idea is that the neural operator learned in a RKBS will be mapping new training points to a new model. The proposed approach makes a clear connection to transformers. The paper presents several compelling applications of the proposed methodology.

The reviewers have asked a number of relevant questions, and I would like to encourage the authors to take them into account to prepare the final version of the manuscript (e.g. relation to other MetaL, computational complexity, etc).

As an additional comment, I would like to encourage the authors to make section 3 less steep and more self contained: for a reader who has not read Zhang (2013), Theorem 1 and Theorem 2 are fairly abstract, and it is difficult to understand what the different mathematical objects are mean to represent in the context of learning without an explanation which would provide some intuition about which kind of object will be learned in which space etc, and a motivation for the structure imposed on these objects. This kind of explanation could probably given with a single additional paragraph before Theorem 1. For the reader not familiar with Zhang, 2013, the relevance of the condition in (3) of Theorem 2 are difficult to digest the way they are presented.